# Impact of interannual and multidecadal trends on methane-climate feedbacks and sensitivity

Chin-Hsien Cheng[1,2] & Simon A. T. Redfern [2,3✉]

We estimate the causal contributions of spatiotemporal changes in temperature ($T$) and precipitation ($Pr$) to changes in Earth's atmospheric methane concentration ($C_{CH4}$) and its isotope ratio $\delta^{13}CH_4$ over the last four decades. We identify oscillations between positive and negative feedbacks, showing that both contribute to increasing $C_{CH4}$. Interannually, increased emissions via positive feedbacks (e.g. wetland emissions and wildfires) with higher land surface air temperature ($LSAT$) are often followed by increasing $C_{CH4}$ due to weakened methane sink via atmospheric $\bullet OH$, via negative feedbacks with lowered sea surface temperatures ($SST$), especially in the tropics. Over decadal time scales, we find alternating rate-limiting factors for methane oxidation: when $C_{CH4}$ is limiting, positive methane-climate feedback via direct oceanic emissions dominates; when $\bullet OH$ is limiting, negative feedback is favoured. Incorporating the interannually increasing $C_{CH4}$ via negative feedbacks gives historical methane-climate feedback sensitivity $\approx 0.08\ W\ m^{-2}\ ^\circ C^{-1}$, much higher than the IPCC AR6 estimate.

[1] Joint International Research Laboratory of Climate and Environment Change, Nanjing University of Information Science and Technology (NUIST), Nanjing 210044, China. [2] Asian School of the Environment, Nanyang Technological University, 50 Nanyang Avenue, Singapore 639798, Singapore. [3] School of Materials Science and Engineering, Nanyang Technological University, 50 Nanyang Avenue, Singapore 639798, Singapore. ✉email: simon.redfern@ntu.edu.sg

Methane is the second most important anthropogenic greenhouse gas (GHG) associated with climate change. The atmospheric concentration of methane was relatively stable in the early-2000s but resumed its earlier growth after 2007 with a further acceleration since 2014[1–7]. The year 2020 marked a milestone of acceleration, despite declining methane emissions from fossil fuel[8], with the subsequent year, 2021, setting another record of increasing $C_{CH4}$[9]. This recent increase of $C_{CH4}$ and a concomitant decrease of the $\delta^{13}CH_4$ isotope ratio have been reviewed[1–7,10] with possible underlying drivers identified as (i) surging biogenic emissions[11–14], (ii) rising fossil fuel emissions with reduced biomass burning[15–17], and (iii) weakening atmospheric and soil methane sink[18–23]. Note that the $\delta^{13}CH_4$ signature of biogenic emissions is lower than that associated with fossil fuel emissions and much lower than the $\delta^{13}CH_4$ signature of biomass burning emissions. Slower oxidation of $CH_4$ leads to lower $\delta^{13}CH_4$ by extending the $^{12}CH_4$ lifetime more than that of $^{13}CH_4$[1–7,23,24]. Biogenic emissions from wetlands and permafrost[5,6,25–33] and atmospheric methane lifetime[10,20–23,34–40] are major methane-climate feedbacks, while other feedback processes, such as wildfires[36,41–44] and natural thermogenic emissions[45], are considered secondary[6,7]. The methane-lifetime feedback contributes the highest uncertainty in estimates of feedback strength[7,40]. Hence, to improve the climate feedback estimates it is essential to constrain past $C_{CH4}$ and $\delta^{13}CH_4$ variations.

In general, warming-induced methane-climate feedbacks tend to be positive, operating through sources, mainly due to accelerated methanogenesis. However, sinks dominate the negative feedbacks with increasing oxidation rate shortening atmospheric methane lifetime, especially due to increases in atmospheric oxidant hydroxyl radicals, $\cdot OH$, from water vapor and lightning-generated $NO_x$[6,7,10,23,27]. Nevertheless, exceptions apply. For example, with respect to LSAT, the atmospheric sink provides secondary positive feedback: higher $CH_4$ emissions, with positive feedback via biomass burning[41,42,44], result in increased atmospheric carbon monoxide (CO), reacting with atmospheric $\cdot OH$ to decrease its concentration, hence extending the $CH_4$ lifetime[22,36,37]. Similarly, increased emissions of biogenic volatile organic compounds (BVOCs) result in positive feedback by limiting $\cdot OH$[40]. For the soil sink (~5% of total sink), accelerated methane oxidation by methanotrophic bacteria may provide a terrestrial methane sink acting as a negative feedback[46], although contradictory observations of a decreasing soil sink, probably due to increased precipitation, have been reported[19]. With respect to precipitation (Pr), high Pr increases wetland extent and water table depth, leading to increased emissions[25,47]. Conversely, low Pr may lead to an increase in forest and peat fires[44,48], resulting in additional associated emissions[36,41,49]. We also note that SST influences the sink via $\cdot OH$ and chlorine. Chlorine forms only a small sink (0.23–2% of tropospheric sink[5,50]) but could potentially increase with rising SST driving increased sea-salt aerosol[51,52]. In addition, SST can influence net terrestrial emissions indirectly via LSAT, Pr, or terrestrial $\cdot OH$. Furthermore, although oceanic $CH_4$ emissions are typically considered minor, with shallow coastal waters being the dominant source[27,53], recent findings on aerobic methane production across the oceans by ubiquitously distributed cyanobacteria and phytoplankton[54–56] add to the uncertainty of feedback strengths.

Here, we apply an empirically verified causal analytical method[57] to quantify the varying causal contributions of T and Pr to changes in $C_{CH4}$ and $\delta^{13}CH_4$, with varying feedback signs differentiated to deduce potential underlying feedback processes.

## Results and discussion
### Estimated climate-contributions to $C_{CH4}$ and $\delta^{13}CH_4$ change.
A material balance helps differentiate the climate and non-climate-contributions (abbreviated as c- and nc-contributions) to changing $C_{CH4}$ and $\delta^{13}CH_4$, as well as different feedback signs among various feedback processes:

$$\frac{dC_{CH4}}{dt} = Q(T, Pr, Q_{nc}) - \frac{C_{CH4}}{\tau(T, Pr, C_{CH4})} = NQ(T, Pr, Q_{nc}, C_{CH4}) \quad (1)$$

$$\sigma \frac{dC_{CH4}}{dt} + (1-\sigma)\frac{dC_{CH4}}{dt} = NQ_c(T, Pr) + NQ_{nc}(Q_{nc}, C_{CH4})$$
$$\approx \left\{ Q(T - \overline{T}, Pr - \overline{Pr}) - \overline{C}_{CH4}\left[ \frac{1}{\tau(T, Pr)} - \frac{1}{\overline{\tau}} \right] \right\}_c \quad (2)$$
$$+ \left[ Q(\overline{Nat}, \overline{Anth}) - \frac{C_{CH4}}{\overline{\tau}} \right]_{nc}$$

where Q represents the monthly emissions, as a function of T, Pr, and non-climate-driven emissions ($Q_{nc}$), $\tau$ is the methane lifetime determined by the flow to total sinks as a function of T, Pr, and $C_{CH4}$. The net monthly emission, or imbalance between all sources and sinks, is then "NQ". Equation 2 differentiates the c- and nc-contributions, with fractions $\sigma$ and 1-$\sigma$. It also indicates the changes in methane production and oxidation rates that result in c- and nc- contributions from the mean state: $Q(\overline{Nat}, \overline{Anth})$ representing the mean natural (Nat) and anthropogenic (Anth) emissions and $-C_{CH4}/\overline{\tau}$ representing the sink with first-order negative-concentration feedback with mean methane lifetime[58].

To quantify the monthly causal contributions from T (or Pr) to $C_{CH4}$, the practically normalized information flow, $nIF_a$, has been estimated, with sign adjusted by their covariance on interannual time scales to differentiate positive and negative feedbacks (see Methods):

$$\frac{\partial C_{CH4}(T)}{\partial t} = \alpha_T \times nIF_{a,T} \times \frac{dT}{dt} \quad (3)$$

$$\frac{\partial C_{CH4}(Pr)}{\partial t} = \alpha_{Pr} \times nIF_{a,Pr} \times \frac{dPr}{dt} \quad (4)$$

where $\alpha$ is a constant calibration factor representing the maximal instantaneous causal sensitivity, obtained by equating the highest peak in observed $dC_{CH4}/dt$ to the estimated total c-contributions, $\partial C_{CH4}(T\&Pr)/\partial t$, i.e., the sum of Eqs. 3 and 4. The partial derivative indicates the "partial" contribution to $C_{CH4}$ as a function of varying T, Pr, or both. Similarly, we estimate the c-contributions to the reconstructed $\delta^{13}CH_4$. In doing so, we suggest that the "normalized causal sensitivity of Y on X" (left-hand side of Eq. 5) is measured by a practical $|nIF_{X\to Y}|$, the fractional uncertainty (or entropy) from the cause-variable (i.e., $IF_{X\to Y}$) over the overall uncertainty perceived by effect-variable Y (i.e., $IF_{(X,non-X,Y)\to Y}$)[57] (see Methods).

$$\left| \frac{\partial Y(X)/\partial t}{dX/dt} \right| \div \left| \max\left( \frac{\partial Y(X)/\partial t}{dX/dt} \right) \right| = |nIF_{X\to Y}| = \frac{|IF_{X\to Y}|}{|IF_{(X,nonX,Y)\to Y}|} \quad (5)$$

Figure 1 shows a comparison of the reconstructed observation of $dC_{CH4}/dt$ (Fig. 1a), $d(\delta^{13}CH_4)/dt$ (Fig. 1i), and their respective estimated contributions from climate feedbacks. The $nIF_a$ between the zonal mean $C_{CH4}$ (or $\delta^{13}CH_4$) and gridded T and Pr has been determined to optimize the estimated c-contributions at each latitude. Our causal analysis is, however, unable to separate the SST-contributions into those that are a direct oceanic influence and those that are an indirect influence through LSAT, Pr, and terrestrial $\cdot OH$. Hence, we apply two different approaches: (i) exclusive means which assume negligible direct oceanic influences to the $C_{CH4}$ or $\delta^{13}CH_4$ (Fig. 1b–e, j–m) and (ii) area-means which assume negligible indirect influence between SST- and terrestrial contributions (Fig. 1f, n). For the first approach, the SST or LSAT+Pr contribution is estimated based

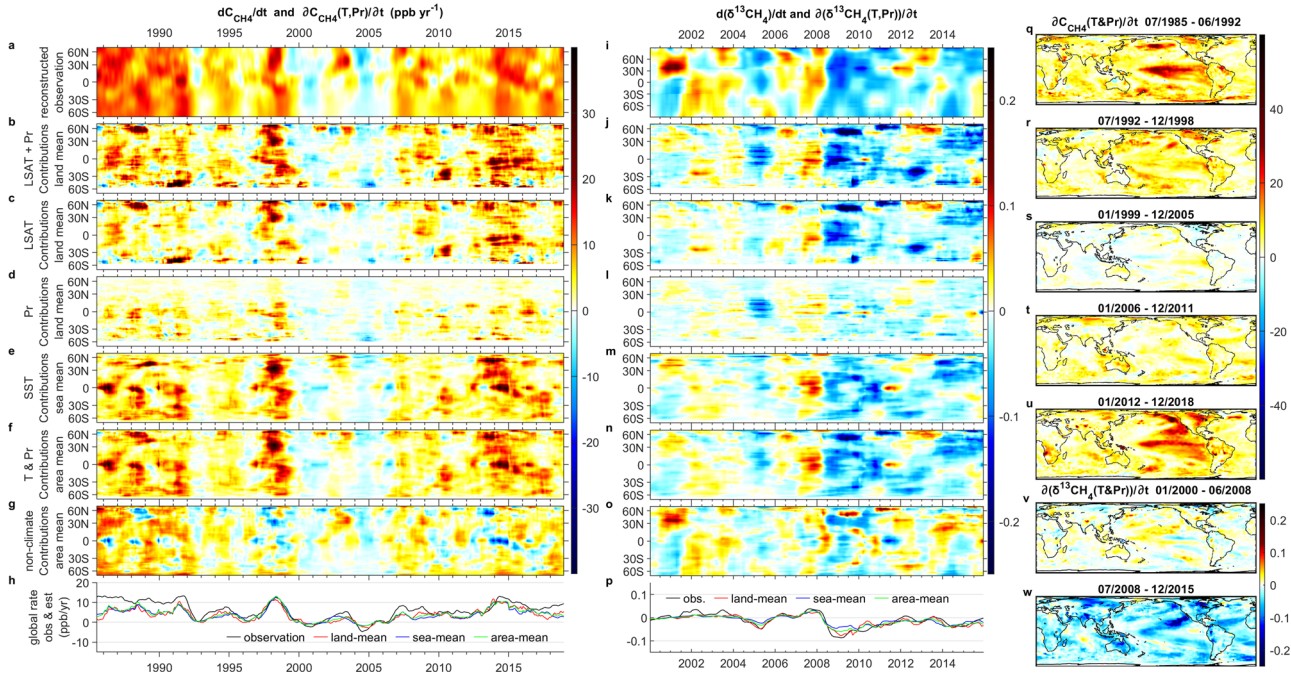

**Fig. 1 Reconstructed observation (obs) of $dC_{CH4}/dt$ (ppb yr$^{-1}$), $d(\delta^{13}CH_4)/dt$ (‰ yr$^{-1}$) and respective estimated (est) c- and nc- contributions.**
**a** $dC_{CH4}/dt$ (ppb yr$^{-1}$) and **i** $d(\delta^{13}CH_4)/dt$ for observations. (est) c- and nc- contributions: **b–f**, **h**, **p–u** for $\partial C_{CH4}(T,Pr)/\partial t$ and **i–n**, **p**, **v–w** for $\partial(\delta^{13}CH_4(T,Pr))/\partial t$. **a–g**, **i–o** *zonal mean* versus latitude (equal area) and time (years CE): *exclusive land* (**b–d**, **j–l**), *sea* (**e**, **m**), or *area-weighted mean* (**f**, **n** for T&Pr and **g**, **o** for nc-contributions); **h**, **p** *global mean*; and **q–w** *temporal mean*. Since the timings of estimated climate-contributions refer to causes that lead to the observed consequences, their difference (**g**, **o**) still shows substantial interannual patterns.

on the exclusive land-mean or sea-mean at each specific latitude, respectively. They should not be summed as both are assumed to fully represent the c-contributions. For the second approach, the contribution from each grid can be considered proportional to the net methane flux, hence the zonal mean contribution is based on the land- and sea-area-weighted mean where fractional land- and sea-areas sum to one at the zonal level. The area-mean appears to best capture the varying meridional distributions although the exclusive land- and sea-means also appear to reasonably reflect the observed interannual variations. This implies that both the direct land- and sea- contributions as well as the indirect contributions via *SST-LSAT&Pr* could be important. Thus we subtract the area-mean from the observed trends to estimate the nc-contributions (Fig. 1g, o). Subfigures 1h and p compare the global means of observed trends and estimated c-contributions, with the difference between them being the nc-contributions. Based on the variations of estimated global mean climate-driven contributions, five periods are classified for $C_{CH4}$ and two periods for $\delta^{13}CH_4$, and the corresponding spatial 2D distributions of temporal means, $\Delta C_{CH4}(T\&Pr)/\Delta t$ and $\Delta(\delta^{13}CH_4(T\&Pr))/\Delta t$, are also shown in Fig. 1q–u and v–w, respectively. The different assumptions for Fig. 1b–h, j–p are irrelevant here since these 2D maps are derived from the mean across temporal dimensions instead of spatial dimension(s). Since positive contributions of $C_{CH4}$ via increasing biogenic emissions and slower oxidation of $CH_4$ lead to a lower $\delta^{13}CH_4$, seen in the climate-driven feedbacks through the major natural source (wetland) and sink (•OH), a positive $\partial C_{CH4}(T\&Pr)/\partial t$ (red) is often accompanied by a negative $\partial(\delta^{13}CH_4(T\&Pr))/\partial t$ (blue), although exceptions may occur.

From the 1980s to the early-2000s the reduction in $dC_{CH4}/dt$ was mainly due to nc-contributions in the northern hemisphere (Fig. 1g). This can be explained principally by the reduced anthropogenic emissions from oil and gas exploitation (especially in the northern hemisphere) and an increasing sink due to higher

$C_{CH4}$[6,58,59] (Eq. 2). The reduction of c-contributions was also substantial (except for the peak during 1997–1998) and it can be explained by the increasing sink due to •OH concentrations[22,23], especially from Southern Ocean warming which considerably strengthened the •OH sink in the southern hemisphere. The resumption of the growth of nc-contributions since 2007 can be interpreted as a result of growing anthropogenic emissions[11–17]. However, likely due to the $-C_{CH4}/\bar{\tau}$ concentration feedback, nc-contributions eventually decrease (the gap between the observation and estimated c-contributions decreases after 2009, Fig. 1h). Unless anthropogenic emissions keep rising at a pace faster than the increasing sink, the nc-contributions would be expected to flatten off again and even trend negatively in the case of declining anthropogenic emissions. On the other hand, the two major peaks of observed $dC_{CH4}/dt$ during 1997–1998 and 2013–2016 are well represented by the estimates from $\partial C_{CH4}(T\&Pr)/\partial t$.

The results for $d(\delta^{13}CH_4)/dt$ shows that the c-contributions capture the negative trends better than the positive trends, suggesting the important role of climate feedbacks in lowering $\delta^{13}CH_4$. Two peaks in the observed $d(\delta^{13}CH_4)/dt$ (above the values given by the means) are seen in the early-2000s, suggesting that the rising $\delta^{13}CH_4$ from fossil fuel emissions over these periods is somewhat independent of climate feedbacks. From 2007 onwards, $d(\delta^{13}CH_4)/dt$ matches closely $\partial(\delta^{13}CH_4(T\&Pr))/\partial t$. This could be a result of roughly equal increases in fossil fuel and agricultural emissions acting to negate the nc-contributions to $d(\delta^{13}CH_4)/dt$[60]. Nevertheless, after 2011, nc-contributions appear to be positive across the northern hemisphere, especially around 30°N, but negative for the tropical and southern hemispheres (Fig. 1o). The positive trend in the north could be explained by increasing coal emissions, especially in China, and the negative trend in the tropics can be explained by increased emissions from the expanding agriculture sector[60].

The estimated $\partial C_{CH4}(T\&Pr)/\partial t$ and $\partial(\delta^{13}CH_4(T\&Pr))/\partial t$ not only match the observed trends well over latitude and time but

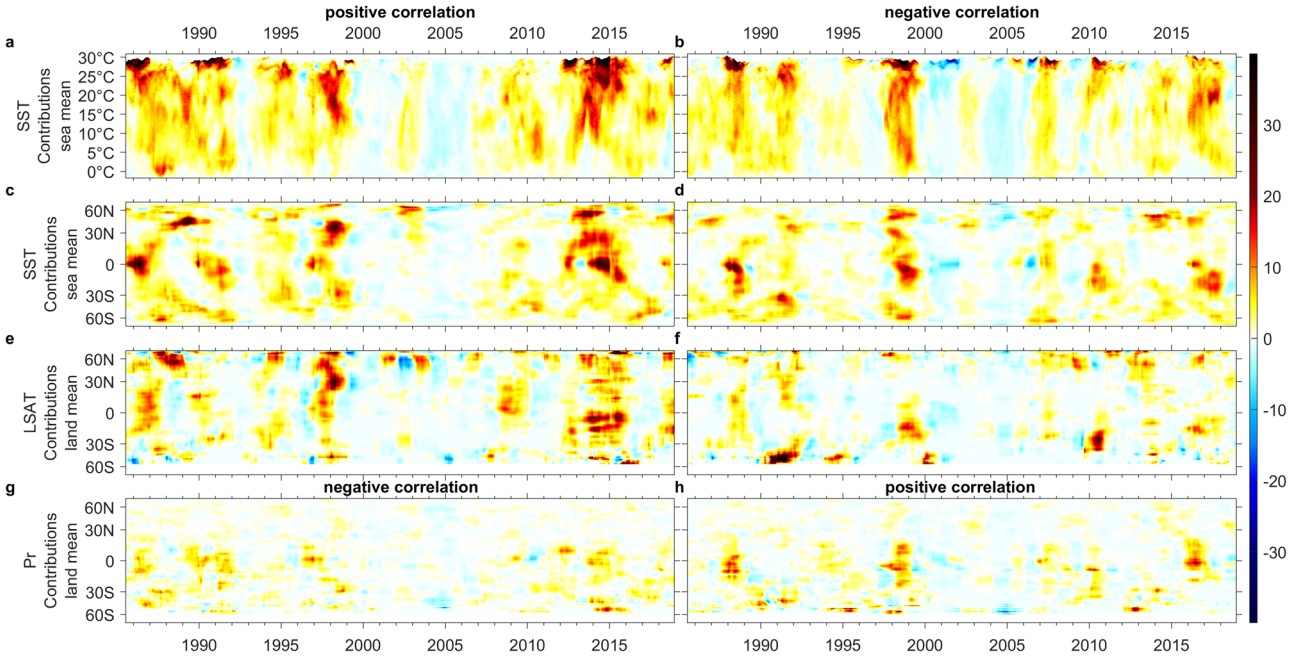

**Fig. 2 Breakdown of estimated exclusive *SST*, *LSAT*, and *Pr* contributions to $dC_{CH4}/dt$ (ppb yr$^{-1}$) with a positive and negative sign of $nIF_a$ (or correlation). c–h** Lat x time coordinate and **a, b** *SST* x time coordinate. A positive $\partial C_{CH4}(T)/\partial t$ could be a result of rising temperature with positive $nIF_{a,T}$ or decreasing temperature with negative $nIF_{a,T}$. Similarly, a positive $\partial C_{CH4}(Pr)/\partial t$ could be due to increasing precipitation with positive $nIF_{a,Pr}$ or decreasing precipitation with negative $nIF_{a,Pr}$.

also spatially (longitude by latitude). North American and Russian wetlands have been identified as the major natural sources of increase at high northern latitudes during 2000–2015[33] (Fig. 1t, u, w). In addition, eastern Russian uplands around Lake Baikal (Fig. 1u–w) also contain considerable amounts of soil carbon[61] and experienced sharp temperature rises. At low latitudes, the substantial contributions from Southeast Asia (e.g., Papua New Guinea and Borneo) and the northwestern regions of Latin America (e.g., Columbia and Equador) are consistent with the high soil organic carbon in these areas[61], while large emissions from trees on the Amazon floodplain[62] are also captured (Fig. 1u). Figure 1w further highlights strong biogenic emissions with negative $\partial(\delta^{13}CH_4(T\&Pr))/\partial t$ from South Asian paddy fields[5,63]. This suggests that the estimates of $\partial(\delta^{13}CH_4(T\&Pr))/\partial t$ are likely more reliable than estimates of $\partial C_{CH4}(T\&Pr)/\partial t$. In particular, rising $C_{CH4}$ may result in underestimates of the negative $\partial C_{CH4}(T\&Pr)/\partial t$, but this is less of a problem for $\partial(\delta^{13}CH_4(T\&Pr))/\partial t$ since it has lower fractional $nc$-contributions and opposite (canceling) trends between fossil fuel and biogenic emissions. See Supplementary Information for examples illustrating the pros and cons between estimated $\partial C_{CH4}(T\&Pr)/\partial t$ and $\partial(\delta^{13}CH_4(T\&Pr))/\partial t$.

**Interannual oscillations of dominant feedback.** Figures 2–5 and Supplementary Figs. 1–3 differentiate the contributions from positive and negative $nIF_a$ (Eqs. 3–4, dependent on the sign of correlation) which can be used to identify the dominant feedbacks. In Figs. 2–3, estimates by the exclusive sea- and land-means are presented as $c$-contributions x lat x time, with land-means broken down into *LSAT*- and *Pr*- contributions. We further present the *SST* contributions as $c$-contributions × *SST* × time, with the aim of differentiating underlying processes, such as direct vs. indirect contributions.

For $\partial C_{CH4}(T,Pr)/\partial t$ (Figs. 2, 4), *SST*- and *LSAT*- contributions with positive correlation and *Pr*- contributions with negative

correlation are contemporaneous, as are *SST*- and *LSAT*-contributions with negative correlation and *Pr*- contributions with positive correlation. A similar contemporaneous pattern can be seen (Figs. 3, 5) for $\partial(\delta^{13}CH_4(T\&Pr))/\partial t$. This implies interferences exist between the oceanic and terrestrial contributions. However, this could be a result of the influence either of *SST* on *LSAT* and *Pr*, or of *SST* on terrestrial •OH concentrations, or possibly of both. In addition, the $c$-contributions switch between positive and negative correlations on an interannual scale (Figs. 2, 3). Typically, periods with strong $c$-contributions from warming-drying and cooling-wetting trends alternate.

This interannually alternating pattern of *SST*- and *Pr*-contributions to $\partial C_{CH4}(T,Pr)/\partial t$ (Fig. 2 and Supplementary Figs. 1, 2) is most obvious in the tropics, while for *LSAT*-contributions such oscillation is spread more evenly across different latitudes. Additionally, from 1998 to 2010 (and possibly 1987–2010), the tropical *SST*- and *Pr*- contributions from cooling-wetting appear to be stronger than those from warming-drying. This is somewhat consistent with the suggested negative correlation between tropical methane emissions and the El Niño Southern Oscillation (ENSO)[25,47]. However, since wetland emissions are supposed to show positive feedback with *LSAT* (i.e., positive correlation with ENSO), the switching pattern of positive and negative correlations, on an interannual scale, more likely can be explained in the following terms:

i. **negative** contributions during cool years through positive *LSAT*-feedback with reduced microbial methane emissions and perhaps the positive *SST*-feedback with strengthened •OH sink (e.g., 1996) →

ii. **negative** contributions through negative *T*-feedback with increased sinks and reduced emissions from anaerobic digestion during warming-drying years with lower water levels (e.g., 1997 in southern hemisphere), or **positive** contributions through positive feedbacks with increased *LSAT*-enhanced wetland emissions and drought (*SST*- and

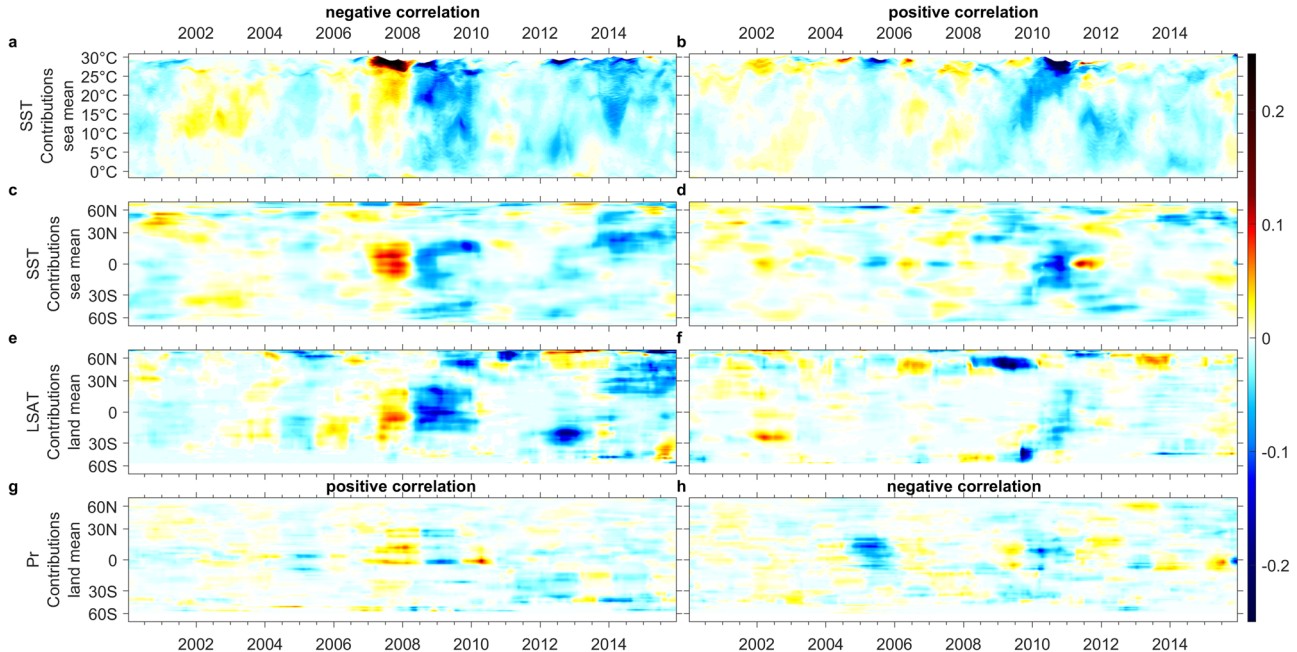

**Fig. 3 Breakdown of estimated exclusive *SST*, *LSAT*, and *Pr* contributions to $d(\delta^{13}CH_4)/dt$ (‰ yr$^{-1}$) with a negative and positive sign of $nIF_a$ (or correlation).** c–h Lat x time coordinate) and **a**, **b** *SST* x time coordinate. A negative $\partial(\delta^{13}CH_4(T))/\partial t$ could be a result of rising temperature with negative $nIF_{a,T}$ (i.e., positive feedback) or decreasing temperature with positive $nIF_{a,T}$ (i.e., negative feedback). Similarly, a positive $\partial(\delta^{13}CH_4(Pr))/\partial t$ could be due to decreasing precipitation with negative $nIF_{a,Pr}$ or increasing precipitation with positive $nIF_{a,Pr}$.

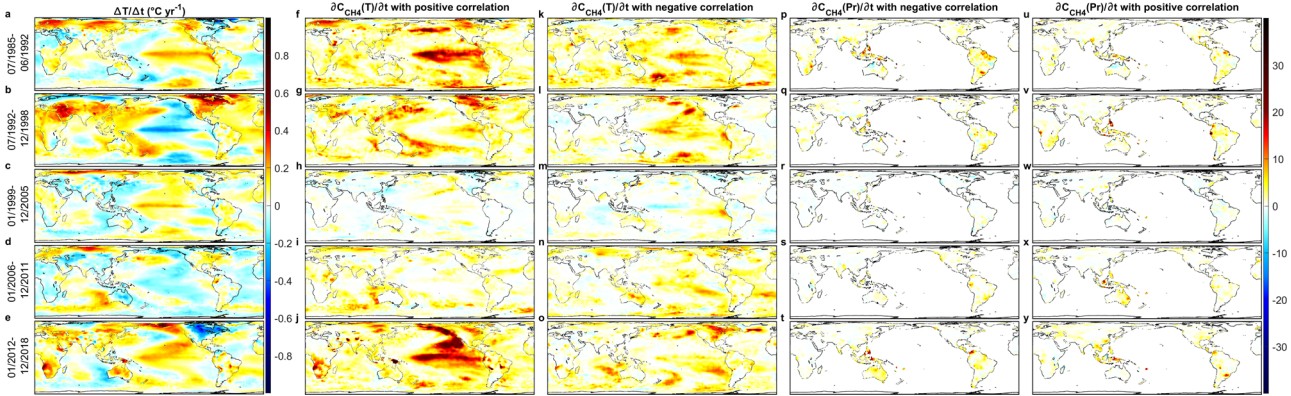

**Fig. 4 Observed spatial distribution of temperature changes and estimated *c*-contributions to $dC_{CH4}/dt$ over different time periods.** **a–e** Observed temperature change (°C yr$^{-1}$). **f–y** Estimated contributions (ppb yr$^{-1}$) from temperature (**f–o**) and precipitation (**p–y**) with positive (**f–j**, **u–y**) and negative (**k–t**) $nIF_a$. The five periods chosen are identical to Fig. 1q–u.

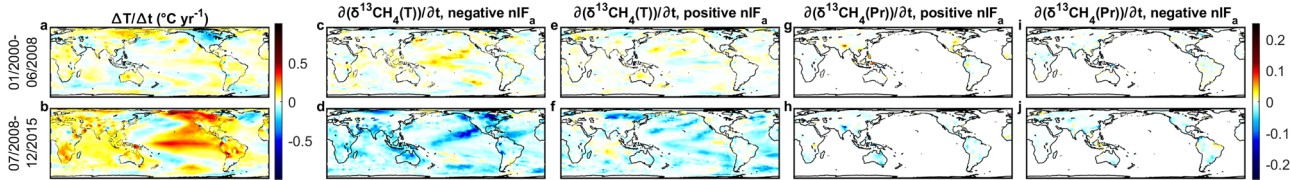

**Fig. 5 Observed spatial distribution of temperature changes and estimated *c*-contributions to $d(\delta^{13}CH_4)/dt$ over different time periods.** a, b Observed temperature change (°C yr$^{-1}$). c–j Estimated contributions (‰ yr$^{-1}$) from temperature (**c–f**) and precipitation (**g–j**) with negative (**c**, **d**, **i**, **j**) and positive (**e–h**) $nIF_a$. Note that negative $nIF_a$ for temperatures more often refers to positive feedbacks through biogenic emissions. Two periods chosen are identical to Fig. 1v–w.

*Pr*)-induced fire (e.g., 1997–1998 and 2013–2015), emitting CH$_4$, CO, BVOC[41–44] →

iii. **positive** contributions through secondary positive feedback by reacting with and consuming atmospheric •OH[22,36,37] →

iv. subsequent **positive** contributions through amplified negative feedback with lowering *SST*, leading to reduced sinks with lower •OH and Cl concentrations (e.g., 1998, 2016–2017); and

v. concurrently *Pr*-driven **positive** contributions through rewetting of peatland resulted in higher water levels and methane emissions (e.g., 1998, 2016).

Note that the wildfire feedback suggested in the Intergovernmental Panel on Climate Change Assessment Report 6 (IPCC AR6)[6] may be underestimated since it mainly considers the increased direct emissions, but not the lost opportunity of enhanced methane sink due to the switch from negative to positive feedback associated with process ii (above), the secondary positive feedback process iii, and the subsequent amplification of positive contributions through negative feedback in processes iv and v.

A similar alternating pattern is seen for $\partial(\delta^{13}CH_4(T,Pr))/\partial t$ (Fig. 3 and Supplementary Fig. 3) between 2005 and 2012, especially from 2007 to 2011. For changes in concentration $\partial C_{CH4}(T\&Pr)/\partial t$, positive correlations are associated with positive feedbacks. In contrast, when changing isotopic signals, $\partial(\delta^{13}CH_4(T,Pr))/\partial t$ show positive correlations this is often a result of negative feedbacks. The interannual alternation of positive and negative contributions to $\partial(\delta^{13}CH_4(T\&Pr))/\partial t$ can be explained thus:

a. **Positive** contributions to $\partial(\delta^{13}CH_4(T\&Pr))/\partial t$ with positive feedback and lowering temperatures at tropical/subtropical wetlands and paddy fields, indicating reduced biogenic emissions, e.g., northeastern India, Southeast Asia, and southern China (2007) →

b. **negative** $\partial(\delta^{13}CH_4(T\&Pr))/\partial t$ with positive feedback indicating increased biogenic emissions (2008–2009) →

c. **negative** $\partial(\delta^{13}CH_4(T\&Pr))/\partial t$ with negative feedback or rewetting indicating reduced •OH, Cl, and soil sink, e.g., tropical Pacific, Atlantic, uplands around India-Pakistan and Bhutan-Bangladesh-Myanmar borders (2009–2010).

In the tropics, the often stronger influence of negative *T*-feedback with rewetting over positive *T*-feedback with drying, especially over the ocean, explains the reported negative correlation with ENSO[25,47]. The alternating pattern is most obvious in the tropics which could be because of (I) the strong *SST* oscillation with ENSO, (II) high *SST* and $H_2O$ vapor level that allows larger fluctuation of •OH concentrations and the $CH_4$ oxidation rate, and (III) relatively higher *Pr* than *LSAT* fluctuation so that the influence of *LSAT* appears milder.

For higher latitudes, although the alternating feedback pattern is less obvious, it is still identifiable, for example at ~50–60°N from 2012 to 2016 (Fig. 2). In contrast to the signals seen in the tropics, the contributions from positive *LSAT* and *SST* feedbacks are stronger than the negative *SST* feedback, especially in northern latitudes. This could be due to the lower *SST* and weaker •OH sink and sharper *LSAT* rise that amplifies stronger wetland emission feedback. The continuous *c*-contributions via alternating feedbacks explain the strong $dC_{CH4}/dt$ peak seen from 1997 to 1998 and again in 2020, since both periods experienced intense wildfires followed by a La Niña year. Nevertheless, in 2020, since the methane emissions from fossil fuels were reduced[8], the $\partial C_{CH4}(T\&Pr)/\partial t$ may have approached or even exceeded the observed $dC_{CH4}/dt$. The more obvious interannual pattern for the mid-high latitudes could be the alternating positive and negative contributions from positive *LSAT* feedback for the northern hemisphere (Figs. 2e, 3e), and negative *LSAT* and *SST* feedbacks as well as positively *Pr*-correlated contributions for the southern hemisphere (Fig. 2d, f, h). The larger fractional ocean area in southern hemisphere may have a role in strengthening the influence of negative feedback through •OH and Cl.

**Multidecadal oscillations of dominant feedbacks.** Two stages of lowering $\partial C_{CH4}(T)/\partial t$ with positive feedback are seen, the first from the late-1980s to 1990s, and a further decline after the late-1990s (Fig. 2a, c, e). This is consistent with the suggested growing negative feedbacks with a higher •OH anomaly occurring from the late-1990s to the mid-2000s[4,18,23] and a milder increase during 1992–1998[4,23]. Nevertheless, $\partial C_{CH4}(T)/\partial t$ gradually increased after ~2007 and $\partial(\delta^{13}CH_4(T))/\partial t$ gradually decreased since ~2008, followed by a sharp strengthening of positive feedback since 2012/2013 (Figs. 2a, c, e, 3a, c, e, 4j, 5d). This suggests weakening sinks[18–23] and/or strengthening biogenic sources[11–13]. While strengthening biogenic sources are apparent in wetlands (e.g., Northern America, Russia, Southeast Asia, and the Amazon, Figs. 4j, 5d) and paddy fields (e.g., India[63] in Fig. 5d), strong positive *SST*-feedback (Figs. 4j, 5d) through a weakening •OH sink seems physically unreasonable, although exceptional positive feedbacks through indirect influences on •OH via *LSAT*-BVOC[40] and *Pr*-related lowering soil sink[19] may still operate.

The puzzling strong positive *SST*-feedback is more easily understood for the tropics, with its influence via *LSAT* and *Pr*, as discussed above. The very strong *c*-contributions through positive *SST*-feedback to both $\partial C_{CH4}(SST)/\partial t$ and $\partial(\delta^{13}CH_4(SST))/\partial t$ around 30°N East Pacific during 2013–2015 is unlikely to be a result of the influence on *LSAT* and *Pr* on paddy field emissions in South and Southeast Asia around the same latitudes. Moreover, during this period, the exclusive sea-mean $\partial C_{CH4}(SST)/\partial t$ is higher than that of exclusive land-mean $\partial C_{CH4}(LSAT\&Pr)/\partial t$ in the northern hemisphere (Fig. 2c, e). Figures 2a, b, 3a, b further highlight the oscillating positive and negative *SST*-feedbacks for *SST* between 10–27 °C (excluding the oscillation >27 °C in the tropics) even when the signals came from latitudes ~60°N during 2013–2014. The most consistent explanation would be cyanobacterial bloom direct methane production for which the optimal temperature is 27–37 °C[56,64]. In contrast, $CH_4$ oxidation by methanotrophs is maximized at slightly lower temperatures, 25–35 °C[65,66]. Net positive feedback is hence still possible when *SST* falls below 27 °C. In addition, as *SST* decreases the influence of the •OH sink through $H_2O$ vapor (negative feedback) becomes weaker, so net positive feedback through direct oceanic emissions is more likely. However, at low *SST*, warming generally decreases the productivity of phytoplankton[67,68], so $CH_4$ production by phytoplankton such as *Emiliania huxleyi*[54] would decrease, such that the likelihood of positive *SST*-feedback is greatly reduced when *SST* drops below ~10 °C. This is seen in the predominance of positive feedback at temperatures greater than 10 °C (Fig. 2a and Supplementary Fig. 4).

These ocean biological processes also have indirect influences on terrestrial feedbacks. In general, the exclusive-mean oceanic contributions precede the exclusive-mean terrestrial contributions, which lead to the observed $dC_{CH4}/dt$ (Figs. 1–2). This implies a potential causality from *SST* to terrestrial *c*-contributions. However, whether this is mainly via *SST*'s influence on *LSAT* and *Pr* or terrestrial •OH may be better differentiated by the variability of terrestrial feedback sensitivities. For instance, Fig. 4 shows strong *c*-contributions through positive wetland feedback in North America during 2012–2018 but the *LSAT* decreased overall during this period, with sharp *c*-contributions through positive feedback only occurring in 2014 and 2015 (see Supplementary Figs. 1–3). In contrast, from 2006 to 2011, North America warmed slightly, but the estimated *c*-contributions are much lower. Earlier in the 1990s (Fig. 4a, b, f, g), increased emissions from North American wetlands are mainly the result of positive feedback. This decadal nonlinearity highlights the need for multidecadal study at monthly (or shorter) resolution. More importantly, such variability of feedback sensitivities is best explained by the various pre-existing conditions of multiple feedback processes. The terrestrial •OH concentration could be the key pre-existing condition determining the net feedback

strength from $LSAT$ and $Pr$. The terrestrial $^•$OH, however, is influenced not only by $SST$-$H_2$O vapor feedback, but also by how much $^•$OH reacts with $CH_4$ emitted directly from the ocean, with remaining $^•$OH reaching the land to influence the feedback sensitivity there. This better explains the roughly contemporaneous pattern of oceanic and terrestrial contributions at higher latitudes, as compared to the indirect influence of $SST$ via $LSAT$ and $Pr$ which does not explain the change in feedback strength.

The multidecadal variability of methane-climate feedback can be explained as an extension of the interannually oscillating feedbacks, while both are hypothesized to be driven by the oscillating dominance between the methane-climate feedback and the methane-concentration feedback (via the -$C_{CH4}/\bar{\tau}$ term in Eqs. 1–2). The rate-limiting factor to methane oxidation could be either atmospheric $^•$OH or the $C_{CH4}$ itself. When the $C_{CH4}$ is a stronger rate-limiting factor, processes that raise $C_{CH4}$ (i.e., net positive methane-climate feedbacks) are more favored. With increasing $C_{CH4}$, the $^•$OH concentration will gradually become more rate-limiting, so the net methane-climate feedbacks will gradually shift from positive to negative. As can be seen, during 1986–1987 (Fig. 2a, b) methane oxidation could be highly $CH_4$-rate-limiting so that positive c-contributions from positive $SST$-feedback could even occasionally occur at ~0 °C. Later, in 1988, an interannual oscillation is seen with positive contributions via negative $SST$-feedback due to more limited $^•$OH concentrations. The interannual oscillation may repeat a few cycles with decreasing •OH concentration (e.g., before 1998), while the range of positive $SST$-feedback in such oscillation gradually decreases (Fig. 2a, c), implying a slow shift towards an $^•$OH-limiting decade (1998–2011) with negative $SST$-$H_2$O-•OH feedback dominance (2000–2010). In brief, on the interannual scale, such oscillation can be amplified by positive terrestrial feedbacks, especially wildfires; on the multidecadal scale, the positive feedbacks via direct oceanic emissions appear to amplify the underlying oscillation.

Our hypothesis of direct-oceanic amplified multidecadal methane-climate feedback might be challenged by the observation of low direct oceanic emissions suggested previously[27,53]. However, direct oceanic emissions can be largely masked by negative $^•$OH feedbacks and thus may have remained undetected during 1998–2011, only becoming significant since 2012. The higher uncertainty of the climate feedback strength from the methane-sink than from the methane-source[7] leaves open the question of the balance between these feedbacks. A good process model should incorporate reaction kinetics describing the oscillating dominance between atmospheric $CH_4$ and $^•$OH as the rate-limiting factor, as well as positive and negative feedback processes from both the lands and seas. However, this is beyond the scope of this study.

We would like to highlight the roughly contemporaneous pattern of switching between positive and negative methane-climate feedback and the pattern of multidecadal $SST$ oscillation. This is clearly seen in the pattern of $SST$ variation during the Interdecadal Pacific Oscillation (IPO), with its negative phase suggested as the key driver behind the global warming hiatus during 1998–2012[69]. Not only does this negative IPO phase coincide with a period of negative methane-climate feedback dominance, its switch to a positive phase with a tripole warming pattern in the northern, tropical (El Niño), and southern East Pacific also coincides with the spatial pattern of $\partial C_{CH4}(SST)/\partial t$ from positive feedbacks during 2013–2015 (Fig. 4j and Supplementary Fig. 1). Dominating negative methane-$SST$ feedback helps stabilize the $C_{CH4}$ and its associated radiative forcing and thus the temperature (and vice versa); however, dominating positive methane-$SST$ feedback could be coupled with accelerated warming. This coupled feedback may introduce additional uncertainty to the modeling of both interdecadal $SST$ oscillation and future methane-climate feedbacks.

**Methane-climate feedback sensitivity and variability.** To project future climate, historical global transient methane-climate sensitivity is understood as a function of global mean surface temperature ($GMST$) anomalies (Fig. 6a–b):

$$\frac{\Delta C_{CH4}(\text{climate})}{\Delta GMST} = \frac{\int_{t0}^{t} \left(\frac{\partial C_{CH4}(TPr)}{\partial t}\right) dt}{\int_{t0}^{t} \left(\frac{dGMST}{dt}\right) dt} \tag{6}$$

Here, $\partial C_{CH4}(T\&Pr)/\partial t$ is the global mean c-contribution estimated according to the assumptions discussed above. With a short lifetime of $9.1 \pm 0.9$ years[6] (or just 6.5–8.8 years during 2000–2009) and a much larger annual $CH_4$ sink than the imbalance (~556:13 based on top-down estimates during 2008–2017)[5], most emitted methane will be oxidized over the multi-centennial time frame of global warming. Hence we also determine the methane-climate sensitivity in terms of annual mean c-contributions per °C $GMST$ change ($\Delta GMST$), obtained by dividing Eq. 6 by the time for $GMST$ change, $\Delta t$ (Fig. 6c, d).

The feedback sensitivity, in ppb °C$^{-1}$, before 1994 initially rises under positive feedback dominance, but declines subsequently and appears to stabilize around 200 ppb °C$^{-1}$ (Fig. 6a, b). This approximates ~0.08 W m$^{-2}$ °C$^{-1}$ (ref. [70]) which is about four times greater than the mean net feedback estimate given in IPCC AR6 (~0.05 positive feedback including permafrost and −0.03 negative feedback, giving ~0.02 W m$^{-2}$ °C$^{-1}$) but agrees within uncertainty[7]. The difference could be largely due to the positive $\partial C_{CH4}(T\&Pr)/\partial t$ from negative feedbacks following the years or decades of positive feedback. In fact, several interannual peaks of sensitivity are due to the positive contributions of lowering $GMST$ (i.e., negative feedbacks). If we breakdown our estimated sensitivity into positive and negative feedbacks, we estimate $0.05 + 0.03$ W m$^{-2}$ °C$^{-1}$ rather than $0.05 - 0.03$ W m$^{-2}$ °C$^{-1}$. Since the 200 ppb °C$^{-1}$ long-term sensitivity is even larger than the estimated absolute maximal instantaneous sensitivity in Eq. 5 (i.e., the calibration factor α in Eqs. 3–4) at 125 (ppb yr$^{-1}$)/(°C yr$^{-1}$), the positive contributions from negative feedbacks should be viewed as lagged responses from earlier positive feedbacks due to nonlinearity. We note that the sensitivity is strongest in boreal and tropical regions (Fig. 6a) due to the positive feedbacks with wetland emissions.

If we consider the sensitivity in terms of ppb yr$^{-1}$ °C$^{-1}$ (i.e., the temperature's influence on the net emission rate), it decreases over the entire period of study (Fig. 6c, d), although there is an increase post-2012. This decrease could be associated with the increasing gradient of $H_2$O vapor pressure per °C rise and a strengthening $^•$OH sink. For the entire period, the decreasing trend is also due to the decadal switch of positive-to-negative feedbacks. The post-2012 increase (Fig. 6c) may be due to the strengthening of positive feedbacks on a decadal scale. In other words, over multidecadal scales, the sensitivity on net emission rate will likely start to decrease once the positive contributions from alternating positive-negative feedbacks start to weaken. The decreasing sensitivity vs $GMST$ post-2012 (Fig. 6d) is a result of the increasing $GMST$ on an interannual scale.

We further project the long-term c-contributions at higher $GMST$ extrapolated from sensitivities based over the entire period of study and post-2012 (Fig. 6e, see Methods). The projection extrapolated from the entire period can be regarded as the multidecadal trend of c-contributions for climate-stabilizing decades (weakening positive and strengthening negative feedbacks). The extrapolation based on the post-2012 period reflects c-contributions during accelerated warming decades (strengthening positive feedbacks). The widening gap between these extrapolations implies increased multidecadal variability of c-contributions at higher $GMST$, which may eventually amplify decadal climate variability as well. Physically, this may be partly explained by the larger range

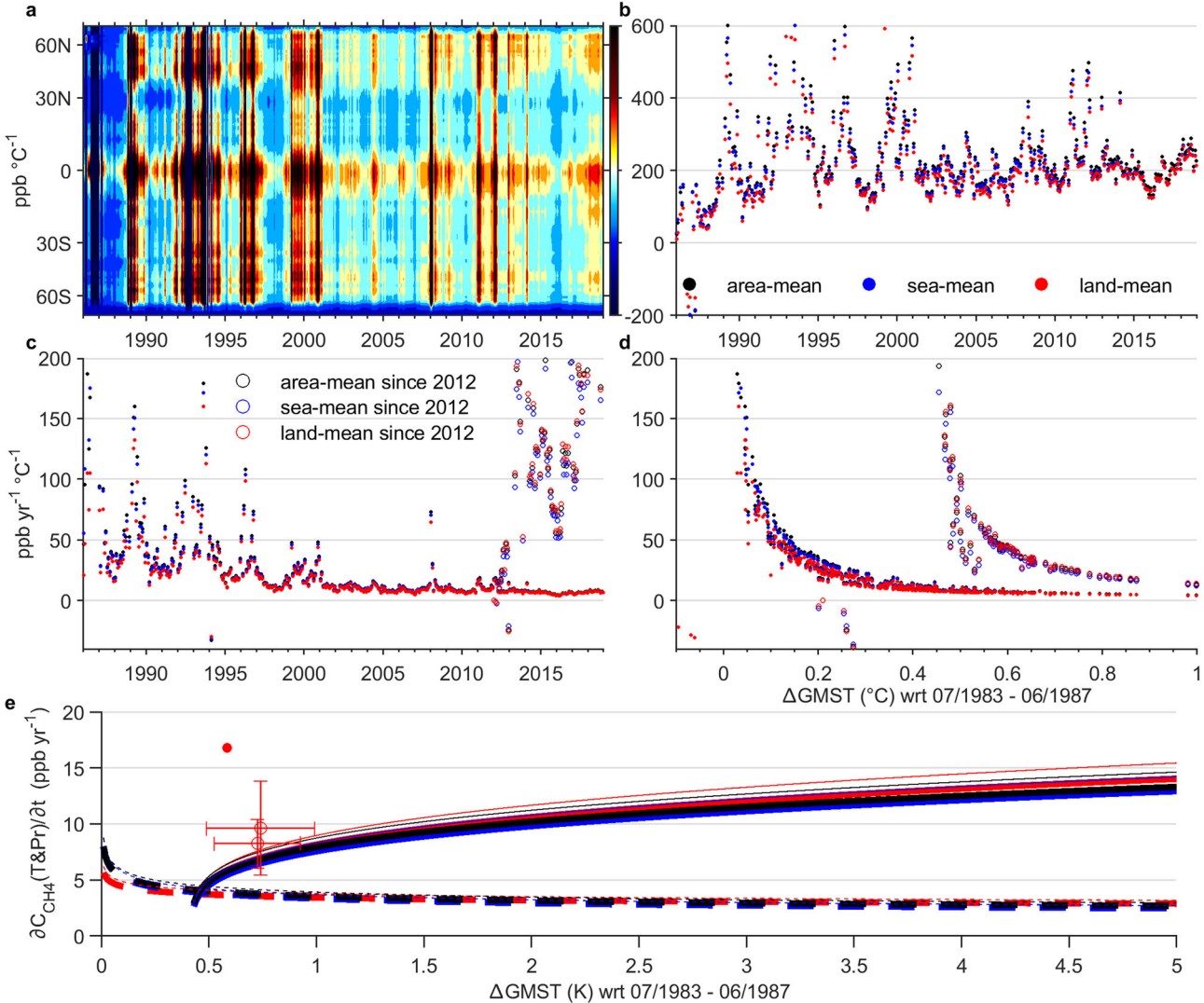

**Fig. 6 Methane-climate feedback sensitivities and projected annual mean climate-feedback-contributions.** Sensitivities are expressed in ppb °C$^{-1}$ (**a**, **b**) or ppb yr$^{-1}$ °C$^{-1}$ (**c**, **d**) vs time (**a–c**), latitudes (**a**), or just °C (**d**). For (**e**), the projected annual mean $\Delta C_{CH4,\ climate}/\Delta t$ curves are fitted based on values with both positive contributions and positive $\Delta GMST$. The projected thick lines are based on extrapolation of global level sensitivities (**d**), and the thin lines are extrapolated similarly but based on a 10% higher calibration factor α. Solid and dashed lines are based on sensitivity trends since 01/2012 and 07/1985, respectively. The two open red circles represent preliminary estimates of $\Delta C_{CH4}/\Delta t$ between 2020 and 2019, and between 2020 and 2018, based on a 75% of imbalance (referring to the σ in Supplementary Table 1). The respective error margins are based on two standard deviations across 12 months of the year. The filled red circle is based on 100% of imbalance between 2020 Dec and 2019 Dec. The 100% assumption is based on the reduced fossil fuel emissions which suggest the observed increase in $C_{CH4}$ could be all driven by climate factors.

of the positive contributions via negative methane-climate feedback during the interannual feedback oscillation at higher temperatures with higher H$_2$O vapor and •OH concentrations. Nevertheless, we are unsure if the historical amplification of variability may be partly associated with increased $C_{CH4}$ (which leads to a larger range of the negative-concentration feedback) and/or ocean eutrophication (which may lead to an accelerated positive feedbacks through cyanobacteria). Furthermore, the observed $dC_{CH4}/dt$ and likely the $\partial C_{CH4}(T\&Pr)/\partial t$ still show increases after 2018, with the preliminary estimates between 2020 and 2018, between 2020 and 2019, and specifically between Dec 2020 and Dec 2019, reflected by two open red circles and one filled red circle, respectively in Fig. 6e. Hence, the upper projected trend line for the accelerated warming decade may yet adjust upward with the incorporation of near-future data.

To summarize, due to nonlinearly lagged responses from positive methane-climate feedback via oscillating positive-

negative feedbacks, the mean value of net methane-climate feedback sensitivity reported in the IPCC AR6 is likely underestimated. The interannual and multidecadal variability of methane-climate feedback may be further amplified at higher temperatures, which may also result in amplified climate variability. However, we are unsure if such increased variability may be mitigated or completely avoided by, for example, a $C_{CH4}$ decrease upon sharp emission cut from anthropogenic sources, or reducing ocean eutrophication. Furthermore, with limited transient historical data, we are unsure how long the lagged positive-feedback responses may last, and whether they outlast a multidecadal oscillation.

## Methods
**Data source**. The reconstructed monthly zonal mean marine surface $C_{CH4}$ were obtained from National Oceanic and Atmospheric Administration (NOAA) Greenhouse Gas Marine Boundary Layer Reference at https://gml.noaa.gov/ccgg/.

The monthly zonal mean surface $\delta^{13}CH_4$ data were reconstructed from 23 surface station datasets of 22 stations obtained from the World Data Center for Greenhouse Gases (WDCGG) at https://gaw.kishou.go.jp/. The $0.5° \times 0.5°$ *LSAT*, $1° \times 1°$ *SST* and precipitation data are based on NOAA Global Historical Climatology Network (GHCN CAMS) Gridded V2, Optimum Interpolation NOAA_OI_SST_V2 and Precipitation Reconstruction over Land (PREC/L), respectively, provided by the NOAA/OAR/ESRL PSL, Boulder, CO, USA, from the website at https://psl.noaa.gov/. The GMST used for sensitivity estimates is reconstructed from the area-weighted average of above mentioned gridded *LSAT* and *SST*.

**Data processing.** Two steps of regressions were applied to reconstruct the station data of $\delta^{13}CH_4$ into monthly zonal mean matrices. First, we estimate the $\delta^{13}CH_4$ of missing months from available stations, based on the regressed trends from stations at nearby latitudes. Second, we interpolate or extrapolate the trends for 180 latitudes. Please refer to the Data Availability for the reconstructed data with equations in Excel spreadsheet format.

To estimate the *c*-contributions to the zonal $dC_{CH4}/dt$ and $d(\delta^{13}CH_4)/dt$ from the observed 3D *T* and *Pr* based on Eqs. 3–4, each time window comprised 49 months (centered month ±24 months). Since the first available data point of monthly $C_{CH4}$ is in July 1983, and the last data point of $C_{CH4}$ that we used corresponds to December 2020, this window moves monthly from (July 1983–July 1987) to (December 2016–December 2020). The range of centered months is hence between July 1985 and December 2018, representing the entire period in all Figures. Similarly, for $\delta^{13}CH_4$ data during 1998–2017, we are only able to estimate the *c*-contributions during 2000–2015. The 4-year window length was chosen to provide a balance between a sufficient length of time-series for reliable analysis and the capability to capture interannual variations of normalized information flow. Besides, limiting the duration allows a more valid comparison with the material balance in expressions 1 and 2. All raw $C_{CH4}$ and climate data were converted into anomalies with respect to a 37-year mean between January 1984 and December 2020. Similarly, the anomaly of $\delta^{13}CH_4$ data with respect to a 20-year mean (1998–2017) was used. To improve the estimates of interannual variability, seasonal trends were removed. The changing rate of these physical variables each month (M), including $dT/dt$ in °C yr$^{-1}$, $dPr/dt$ in mm day$^{-1}$ yr$^{-1}$, $dC_{CH4}/dt$ in ppb yr$^{-1}$, and $d(\delta^{13}CH_4)/dt$ in ‰ yr$^{-1}$, was derived as the difference between the mean in 1 year forward (M to M+11 months) and the mean in 1 year backward (M-12 to M-1 months). Once the 3D causal contributions were determined, they were folded down to 2D data (lat × time) based on exclusive land- or sea-means or area-weighted means to obtain the zonal mean. Converting the zonal mean to the global mean was performed, accounting for area-weighting across the latitudes.

**Quantifying the causal contributions with normalized information flow.** Information flow (IF) and its normalized form (*n*IF)[71–73] are established measures of causality between two dynamical events realized in the form of, in a typical case, time-series. Empirically, we find that a normalized causal sensitivity between two time-series variables can be approximately described by normalized information flow (Eq. 7)[57]. Such normalization could be applicable to different causes, from various locations and times. Together with the correlation sign (8), its rewritten form can be used to quantify the causal contributions (9).

$$\left|\frac{\frac{\partial Y_X}{\partial t}}{\frac{dX}{dt}}\right| \div \max\left|\frac{\frac{\partial Y_X}{\partial t}}{\frac{dX}{dt}}\right| = \frac{\text{causal sensitivity of } Y \text{ to changing } X}{\text{maximal causal sensitivity of } Y \text{ to changing } X}$$

$$\approx \frac{\text{flow of uncertainty from } X \text{ to } Y}{\text{overall flow of uncertainty to } Y \text{ from } X, \text{non} X, \text{and } Y \text{ itself}}$$

$$= \frac{\left|IF_{(X \to Y)}\right|}{\left|IF_{(X, \text{non}X, Y \to Y)}\right|} = \left|nIF_{(X \to Y)}\right| \quad (7)$$

$$nIF_a(X \to Y) = \left|nIF(X \to Y)\right| \times (\pm 1, \text{based on correlation}) \quad (8)$$

$$\frac{\partial Y_X}{\partial t} = \alpha \times nIF_a(X \to Y) \times \frac{dX}{dt} \quad (9)$$

where the $\alpha$ is a calibration factor representing the maximal causal sensitivity in Eq. 7.

Given two time-series, say, $X$ and $Y$, it has been shown[72] that the maximum likelihood estimator of the information flow from $X$ to $Y$ is given by:

$$IF_{X \to Y} = \frac{C_{YY}C_{YX}C_{X,dY} - C_{YX}^2 C_{Y,dY}}{C_{YY}^2 C_{XX} - C_{YY}C_{YX}^2} \quad (10)$$

where $C_{YX}$ is the covariance between variables $Y$ and $X$, and $C_{X,dY}$ is the covariance between $X$ and $\dot{Y}$, a series approximating $dY/dt$ using Euler forward differencing scheme ($\dot{Y}_n = (Y_{n+1} - Y_n)/\Delta t$). The same system of notation applies to $C_{XX}$, $C_{YY}$, and $C_{Y,dY}$ too. Here only absolute values of IF are considered since, ideally, a nonzero IF indicates causality.

The normalized information flow[73] is obtained by dividing the IF by a normalizer $Z$ (11). The definition of the normalizer was first proposed by Liang (12)[73]. However, empirical assessment[57] suggests an alternate normalizer (13) to better reflect the *n*IF defined in Eq. 7.

$$nIF_{X \to Y} = \left|IF_{X \to Y}\right|/Z_{X \to Y} \quad (11)$$

$$Z_{X \to Y} = \left|IF_{X \to Y}\right| + \left|\frac{dH_Y^*}{dt}\right| + \left|\frac{dH_Y^{\text{noise}}}{dt}\right| \quad (12)$$

where $\left|\frac{dH_Y^*}{dt}\right| + \left|\frac{dH_Y^{\text{noise}}}{dt}\right|$ is the estimated increase in marginal entropy (extent of uncertainty) $H_Y$, which includes the rate of change of $H_Y$ due to $Y$ itself (first term) and the contribution from noise (second term). The $\left|\frac{dH_Y^{\text{noise}}}{dt}\right|$ also corresponds to the $\left|IF_{(\text{non}X \to Y)}\right|$ and $\left|\frac{dH_Y^*}{dt}\right|$ corresponds to the $\left|IF_{(Y \to Y)}\right|$ in Eq. 7.

$$Z_{X \to Y} = \left|IF_{X \to Y}\right| + \left|\frac{dH_Y^{\text{noise}}}{dt}\right| + \left|\left|\frac{dH_Y^*}{dt}\right| - \left|IF_{X \to Y}\right| - \left|\frac{dH_Y^{\text{noise}}}{dt}\right|\right| \quad (13)$$

The key motivation behind the modification is to correct the $\left|IF_{(X, \text{non}X, Y \to Y)}\right|$ in Eq. 7 from the direct sum of the three separate terms into the sum of $\left|IF_{X \to Y}\right|$ and $\left|IF_{(\text{non}X \to Y)}\right|$, together with the additional generation of IF (i.e., $\left|\left|IF_{(Y \to Y)}\right| - \left|IF_{(X \to Y)}\right| - \left|IF_{(\text{non}X \to Y)}\right|\right|$). This correction addresses the intersecting information flow between (i) the flow from $X$ and non-$X$ causes and (ii) the flow received by effect-variabe $Y$. Before the correction, the $\left|IF_{X \to Y}\right| + \left|\frac{dH_Y^{\text{noise}}}{dt}\right|$ actually approaches $\left|\frac{dH_Y^*}{dt}\right|$ when the causal sensitivity is about maximized, resulting in the $|nIF|$ proposed by Liang to approach 0.5 instead of 1[57]. Such correction has been found to minimize the error of Eq. 9 when the estimated correlation sign misinterprets the feedback direction. Our results are based on the definition given in Eq. 13.

Three conditions are suggested for the application of this method:(i) there are strong noise contributions from hard-to-quantified independent sources; (ii) there are significant time-lags between causes and effects, especially when we would like to estimate when the causes have occurred; (iii) there are many sources of causal contributions from various spaces to a common effect, especially when we would like to estimate where the causes are from. For the case of methane-climate feedback, the anthropogenic emissions provides the first condition, while our interest in identifying and differentiating the spatiotemporal variability of methane-climate feedback contributions fulfils the second and third conditions. We have compared the estimates based on different modified normalizers for *n*IF, unnormalized IF and linear regression (i.e., replacing the $nIF_a(X \to Y)$ in Eq. 9 by $IF_a(X \to Y)$ or $mR^2$ between $X$ and $Y$) (Supplementary Fig. 5). The estimates given by the normalized information flow show a clear advantage; while the results based on two different definitions of normalizer only show the marginal difference and do not affect our main discussions.

**Pros and cons between estimated $\partial C_{CH4}(T\&Pr)/\partial t$ and $\partial(\delta^{13}CH_4(T\&Pr))/\partial t$.** As an example, the weak positive $\partial C_{CH4}(T\&Pr)/\partial t$ seen for the tropics in 2007 may be a false signal since it coincides with a strong positive $\partial(\delta^{13}CH_4(T\&Pr))/\partial t$ signal in areas that are better explained in terms of weakened wetland and paddy field emissions (Supplementary Figs. 1–3 for the yearly maps). However, we also note that positive wildfire feedback and biogenic emissions have opposite effects on $\partial(\delta^{13}CH_4(T\&Pr))/\partial t$. For example, the weaker $\partial(\delta^{13}CH_4(T\&Pr))/\partial t$ signals seen in Fig. 1w when compared with $\partial C_{CH4}(T\&Pr)/\partial t$ in Fig. 1u over the tropical Pacific might be explained by the severe Indonesian wildfires in 2015[44]. Unfortunately, the raw data availability for $\delta^{13}CH_4$ are more limited, leading to higher uncertainty in our reconstructed trends.

**The long-term multidecadal projection (Fig. 6e).** While long-term sensitivities in ppb °C$^{-1}$ (Fig. 6a, b) or ppb yr$^{-1}$ °C$^{-1}$ (Fig. 6c, d) are estimated based on Eq. 6 and further division of length of time, procedures for projection of the *c*-contribution in ppb yr$^{-1}$ (Fig. 6e) is based on a linear regression between ln($\Delta GMST$) and the ln(sensitivity in ppb yr$^{-1}$ °C$^{-1}$) with negative values and extreme values filtered off before taking the natural logarithm; followed by extrapolation of the ln(sensitivity in ppb yr$^{-1}$ °C$^{-1}$) and hence the *c*-contribution in ppb yr$^{-1}$ between 0.01–5 °C $\Delta GMST$.

**Caveat.** The most important caveat of the method is the likely underestimate of negative $\partial C_{CH4,T\&Pr}/\partial t$ due to the contributions of anthropogenic emissions, as we have mentioned. In addition, the calibration factor $\alpha$ (i.e., the hypothesized maximal instantaneous causal sensitivity at 125) is assumed identical for all different causes (i.e., *LSAT*, *SST*, and *Pr*) and is changed only for estimating $\partial C_{CH4,T,Pr}/\partial t$ and $\partial(\delta^{13}CH_4)_{T,Pr}/\partial t$. This assumption is applied even with different units of causes (i.e., °C and mm day$^{-1}$). This calibration is rather approximate, by equating the highest peak of observed $dC_{CH4}/dt$ and the $\partial C_{CH4,T\&Pr}/\partial t$ around 1998, and $d(\delta^{13}CH_4)_{T\&Pr}/dt$ and $\partial(\delta^{13}CH_4)_{T\&Pr}/\partial t$ around 2009, both at the zonal and global level (Fig. 1). To a certain extent, the calibration at the zonal level helps build the

case for applying the same $\alpha$ for temperatures and precipitation. We note that the $c$-contributions from *LSAT* and *Pr* overlap to a certain extent, but they are assumed mutually exclusive here when we sum them to estimate the overall territorial causal contributions. Nevertheless, for the uncertainty of the future $c$-contribution projection, the uncertainty due to the method, such as multiplying the $\alpha$ by 1.1 to match the estimated and observed peaks during 2013–2014 instead of the peaks during 1997–1998, is significantly lower than the uncertainty from interdecadal variability (Fig. 6e). Examples of other minor sources of uncertainty include (i) the uncertainty in estimating the $\left|\mathrm{IF}_{T \to C_{\mathrm{CH4}}}\right|$ and the normalizing factor $\left|\mathrm{IF}_{T,\mathrm{non}T,C_{\mathrm{CH4}} \to C_{\mathrm{CH4}}}\right|$ and (ii) the assumed locality of zonal $C_{\mathrm{CH4}}$ for estimating the causal contributions of gridded $T$ and $Pr$ on an interannual scale. For (i), given the short time-series (49 numbers) for estimating the maximal likelihood IFs, we did not estimate the uncertainty range. For (ii), it is considered minor since meridional mixing for $C_{\mathrm{CH4}}$ takes multiple years or even a decade for the $C_{\mathrm{CH4}}$ in the southern hemisphere to catch up the $C_{\mathrm{CH4}}$ in the northern hemisphere. Nevertheless, since our general findings are based on the alternating feedback (correlation) sign and the varying causal sensitivity (which is the core advantage of the method) our results are sufficiently robust.

Some region-specific results should be interpreted carefully. In the southern hemisphere, patterns are seen in unusual areas including the Angolan uplands and Northern Australia. While the Angolan uplands could serve as an important soil methane sink[5] and be affected by temperature, we have not identified a convincing physical explanation for the signals from Australia. Nevertheless, we can compare Fig. 1 with Supplementary Figs. 1–3, which show the maps of yearly $\partial C_{\mathrm{CH4}}(T\&Pr)/\partial t$ and $\partial(\delta^{13}\mathrm{CH4}(T\&Pr))/\partial t$. This comparison reveals contributions to the increasing regional $\partial C_{\mathrm{CH4}}(T\&Pr)/\partial t$ in 2010 and 2016, in Angola and Australia respectively. Until we have reasonable explanations for the observed $dC_{\mathrm{CH4}}(T\&Pr)/dt$ for these specific years and latitudes, we cannot rule out the importance of these results.

**Examining $\partial C_{\mathrm{CH4}}(T\&Pr)/\partial t$ assuming proportionality to $T\&Pr$ instead of $dT/dt$ and $dPr/dt$.** In Eqs. 3 and 4 we apply causal analysis with an assumption that $\partial C_{\mathrm{CH4}}(T\&Pr)/\partial t$ could be proportional to $dT/dt$ and $dPr/dt$. This is in contrast to typical approaches that describe the dependence of emission rate and hence $\partial C_{\mathrm{CH4}}(T\&Pr)/\partial t$ in terms of $T$ or $Pr$. We have, therefore, also calculated the causal contribution based on this traditional assumption (Eqs. 14–15)

$$\frac{\partial^2 C_{\mathrm{CH4}}(T)}{\partial t^2} = \alpha_T \times n\mathrm{IF}_{a,T} \times \frac{dT}{dt} \text{ and } \frac{\partial C_{\mathrm{CH4}}(T)}{\partial t} = \alpha_T \times n\mathrm{IF}_{a,T} \times T \quad (14)$$

$$\frac{\partial^2 C_{\mathrm{CH4}}(\mathrm{Pr})}{\partial t^2} = \alpha_{\mathrm{Pr}} \times n\mathrm{IF}_{a,\mathrm{Pr}} \times \frac{dPr}{dt} \text{ and } \frac{\partial C_{\mathrm{CH4}}(\mathrm{Pr})}{\partial t} = \alpha_{\mathrm{Pr}} \times n\mathrm{IF}_{a,\mathrm{Pr}} \times \mathrm{Pr} \quad (15)$$

As Eqs. 3 and 4 are based on the causal analysis between $C_{\mathrm{CH4}}$ and $T$, $Pr$ time-series, for the assumption behind 14 and 15, the causal analysis is conducted between $dC_{\mathrm{CH4}}/dt$ and $T$, $Pr$ time-series. The results of estimated $\partial^2 C_{\mathrm{CH4}}(T\&Pr)/\partial t^2$, however, deviate from the observed trends significantly; the estimated $\partial C_{\mathrm{CH4}}(T\&Pr)/\partial t)$ (as the integral of estimated $\partial^2 C_{\mathrm{CH4}}(T\&Pr)/\partial t^2$) are generally negative (meaning decreasing $C_{\mathrm{CH4}}$), in contrast to the observed interannual positive-negative variations (Supplementary Fig. 6). This suggests that the assumption of linear proportionality between $\partial C_{\mathrm{CH4}}(T\&Pr)/\partial t$ and $T$ or $Pr$ for the methane-climate feedback variability is inappropriate. Again, this highlights the importance of describing $\partial C_{\mathrm{CH4}}(T\&Pr)/\partial t$ by $dT/dt$ and $dPr/dt$, in order to factor in the nonlinearity due to process hysteresis, as we discuss.

## Data availability

The zonal $\delta^{13}\mathrm{CH4}$ reconstruction on a monthly scale used in this study has been deposited in the figshare database under accession code https://doi.org/10.6084/m9.figshare.19642293.v1.

## Code availability

The Matlab codes used in this study have been deposited in the figshare database under accession code https://doi.org/10.6084/m9.figshare.19642302.v1.

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

## Acknowledgements
An earlier phase of this study was supported by the National Key Research and Development Program of China (2017YFA0603804). This work was supported by NTU Singapore's start-up grant to SATR.

## Author contributions
C.-H.C. conceptualize the project, methodology, data curation, and analysis. C.-H.C. and S.A.T.R. discussed the analysis and results, and prepared the manuscript together.

## Competing interests
The authors declare no competing interests.

## Additional information

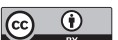

