## [Peer Review File · Nature Communications]

Impact of interannual and multidecadal trends on methane-climate feedbacks and sensitivityREVIEWER COMMENTS

Reviewer #1 (Remarks to the Author):

The authors fit a statistical model based on normalized information flow to estimate the rate of change of CH₄ based on temperature and precipitation. I agree with the authors that a better understanding of changes in methane-climate feedbacks are important and also believe that statistical methods such as information flow, transfer entropy, etc can be very valuable tools in detecting coupling between variables and potential feedbacks, but I am not convinced at this point that the manuscript convincingly elucidates the interplay between positive and negative feedbacks.

Presentation of methods and results

I think that this manuscript is a case in which presentation of methods, figures, and results gets into the way of understanding the central claims of the paper. For example:

- the four included figures present a total of 60(!) subplots, most of which are shaded pseudocolor plots for which it is difficult to discern clear differences.

- axis labels are sometimes missing and figure headings are small. For example, it is necessary to zoom into Figure 2 to see the difference between the b and c column and column headings don't easily translate from variables/ jargon to a physically understandable process.

- In general the paper uses a lot of shorthand jargon, which makes parsing contents difficult. For example (p9, L8)

"Contributions with positive nIF_{c,SST}, positive nIF_{c,LSAT}

and negative nIF_{c,Pr} are contemporaneous as are contributions with negative nIF_{c,SST}, negative nIF_{c,LSAT}, and positive nIF_{c,Pr}. This suggests significant indirect SST-feedbacks through the LSAT and Pr" > If I understand correctly, this sentence could be summarized that there is both positive and negative information flow from temperature and precipitation at the same time. I am not sure how the second part of the sentence then follows from that. My general recommendation would be to reduce the amount of variable names and to at least partially translate these concepts into plain language.

- The way the method section is divided between main text and supplement makes it difficult to understand, what exactly is calculated from what. I feel that instead of having eq1 and eq2 in the paper the paper would be better served with providing a clear and concise description of the basic assumptions of the information flow method as applied in this manuscript including some of the basic limitations. The methods section in the supplement should then contain a clear description of the calculations as they are computed.

Local vs non-local effects and validity of results:

It is my understanding that the authors use local fields of SST/LandT and Precip together with a boundary layer dataset of CH₄ to estimate establish whether changes in T and P inform changes in CH₄. My biggest concerns about this as presented here is the following. T and P are local fields, but - given the long (20 or so year) lifetime of methane and the monthly timestep - the change in methane in each grid cell is a composite of local effect and atmospheric transport and atmospheric mixing within a hemisphere occurs on a two-week timescale. This makes me question whether the resulting information flows present a detectable process locally or "just" the result of the statistical model having to fit the nIF locally. The result of that could be that the nIF basically average to something closer to zero over time, which then may be interpreted to changing feedback. One key piece of information not shown by the author would be the reconstructed field of dCH₄/dt (similar to figure 1a) from their analyses rather than the temporal figures 1m-q, which are not compared to observations (or a comparison of 1m-q to data).

Methods

- Building on the previous point, it is not clear to me how to verify that information flows are statistically significant. Generally speaking establishing significance in information theory methods is not easy and requires for example bootstrapping. I think that this is critically lacking.

- the authors apply a single 'calibration' factor (i.e. scaling factor) of estimated results, to estimate the temperature and precipitation driven change in CH₄. It is not apparent to me whether this is justifiable and the choice of this should be clearly justified in the methods.

- To what extent is the assumption valid that patterns in dCH₄/dt represent climate contributions and not for example time varying changes in anthropogenic emission. I am not saying that this is

the case, but should be addressed in the manuscript.

Diagnosing feedback

- It is not clear to me to what extent this is actually diagnosing feedback. Information flow diagnoses directional coupling between variables, but that is not the same as feedback. For example, the paper diagnoses $IF(T \rightarrow CH_4)$, but a true diagnosis of feedback requires diagnosing the entire loop including $IF(CH_4 \rightarrow T)$ or some other loop. I don't think that positive and negative nIF values can be assumed to be stand-ins for positive and negative feedback. This should be clearly demonstrated or elaborated on.

Reviewer #3 (Remarks to the Author):

The manuscripts quantify the contributions of changes in temperatures and precipitations to the atmospheric methane variability based on historical data. Changes in temperatures and precipitations could induce both positive and negative feedbacks through changes in climate driven emission or sinks (e.g., H_2O-OH or $CO-OH$). Events with different dominant feedbacks are discussed. The combined feedbacks have been shown to limit the climate driven methane emissions and its sensitivity to climate-feedback in the future. The results highlight the significance of climate-driven contributions to the methane variability. The manuscript is well written but the methodology needs additional clarification. The reviewer therefore suggests a minor revision.

General comments:

Changes in temperature and precipitation induced impacts on wetland emissions are discussed in the manuscripts. How about the induced impacts on anthropogenic emissions? For example, temperature and precipitation would also affect rice paddies, which is a large source of methane, especially over tropics. It is worthy to include related discussion in the manuscript. On the other hand, the combined contributions from changes in temperatures and precipitations are likely to be biased to some extent in this study, as in many cases, changes in temperatures would affect water cycle and therefore precipitation. Also, it would be more straightforward if we could have a more quantitatively estimates in the land/sst contributions.

Specific comments:

Page 3, lines 21-24, what about their impacts on rice paddies?

Page 4, line 14, should be "as well as positive feedbacks"?

Page 4, line 16, how does this compare to your SST-contribution estimated in this work?

Page 5, lines 5-10, how do you define anthropogenic activity here? Do you assume they are not affected by climate variables? But in fact, climate change would affect anthropogenic emissions. Also, how do you determine σ ?

Page 7, Figure 1, in Fig1f, if I understand correctly, the observation is based on global mean, the land contribution is based on land mean, and the SST contribution is based on sea mean. So what does the combination of the land contribution and SST contribution suggest here? If we add land-contribution and sea-contribution together, it is much larger than the observations. I do not think anthropogenic emissions have negative contribution here. Similar for Fig.1l. What is a better way to interpret this? On the other hand, is it possible to show the contribution in a more quantitative way, say in %?

Page 8, lines 1-5, how do you get this? Figure 1 only shows the climate-drive contribution, which gradually decreases from 1980s to mid2000s. Also, from Eqn (2) for Q_{nc} term, besides the increasing sink, how about the trend in $Q(nat,ant)$? Is it possible to quantitatively compare the contributions from Q_{nc} and QC ?

Page8, lines6-10, "But these appear to eventually diminish... ", based on what?

Page 12, lines 1-3, what about agricultural emissions?

Page 14, lines 1-8, Doesn't it mean due to reduced OH levels from intensive wildfires, it tends to increase CH_4 and therefore contributes to a peak in 2020? I do not think a negative SST- H_2O-OH feedback itself could have such big impact on the increase in CH_4 . Also, the 1997-1998 ENSO is much stronger than 2019-2020 but the changes in CH_4 are smaller than 2019-2020.

Page 15, line 18-19, does this mean methane variability would not be sensitivity to future climate change? In other words, non-climate contribution would play a role in the future?

Page 17, Figure 4, why do you set the same fractional area (i.e., 0.25) for both northern and southern mid/high latitudes? Is it more reasonable to have a higher fractional area in the Northern Hemisphere? Where is table 1? There is no filled red circle in Fig4i.
Page18, lines 5-8, can CMIP6 future projections provide some constraints on the estimates here?

Supplement:

Page5, line9, should be "SST-driven"

Page5, lines 20-22, does this mean the sum could lead to overestimates in the overall territorial contribution?

Reviewer #4 (Remarks to the Author):

Comments on NCOMMS
MethClimateFeedback313740_0

General

This is a very interesting pioneer paper. I'm not sure I agree with some of its suppositions, and I think some of the inputs are either dated or wrong, but the paper is very innovative and definitely interesting, and potentially offers a major new route to solving an important question. Thus I support publication after some revision.

Arguably the most interesting question in studying atmospheric methane is not why it is rising at record growth rate right now (which is an extremely interesting puzzle), but the even more important puzzle whether the warming is feeding the warming. We all have strong suspicions that warming is indeed driving methane emission and hence driving warming, but these are largely gut feelings: it is very tough to say this specific emission rise comes from this specific temperature rise.

This paper addresses that specific problem – using Liang's 'normalised information flow' approach, it develops a methodology to find causal connections.

Now I need to caution that casual coincidence is NOT proof of causal connection, but it's interesting, really interesting.....

That said, I have a number of specific quibbles with the paper. In particular, the peatland information is very badly out of date (page 12) and the lack of mention of the tropical ruminant source and the Cl sink both need to be considered.

Note, there is barely any use of isotopic information.

Thus I recommend publication after revision.

Specific

Please could Nature Comms send out its review documents with Line Numbers continuously numbered through the text pages....it is so much easier on the referee and so easy to do!!

Abstract: line 6-7 "SST related OH likely" – rewrite as the English is ugly and ambiguous.

Page 4 line 7-9. No mention here of the Cl sink. I know it is small but it is significant when discussing ocean emissions, and it is latitudinally distributed. It should be included. Various papers on this - see for example Hossaini, R., et al. (2016) A global model of tropospheric chlorine chemistry: Organic versus inorganic sources and impact on methane oxidation. Journal of Geophysical Research: Atmospheres 121.23 (2016).

Page 4 line 14. The problem of ENSO and wetlands is not simple. First, both heat and water are involved. Because methane emission is exponential with Temperature – Arrhenius relation – then a hot but fairly dry wetland can in principle emit more methane than a cool wet wetland. Secondly there is time factor – groundwater is important. If the previous season was wet, then a dry El Nino season may still produce enough run off to produce a soggy wetland, while conversely during a wet La Nina year the run off may be so consumed in the task of raising groundwater levels that there is not much expansion of wetland area.

Page 4 line 17 – this is the only mention of isotopes, yet surely isotopes are central to solving the anthropogenic vs natural puzzle! That’s arguably the big weakness of the whole paper.

Page 5 line 12 – Normalised information flow. I think this concept needs an introductory paragraph. Yes, readers can read Liang’s papers, but it would help to introduce the concept in the text here.

Page 6 lines 2 and 3. In this context it might be worth mentioning the NOAA Sine-Latitude-Time plots....see Growth Plot in: <https://gml.noaa.gov/ccgg/figures/> The plot is also used in the Nisbet et al. paper.

Page 6 line 20. Careful – wetlands ARE important but so also are tropical cows (see Schaefer et al 2016, and both Nisbet et al papers). Moreover cows and wetlands are very similar – similar methanogens in the grass-to-methane factory, same link to rainfall (more rain, more fat cows), same isotopic signatures, and similar latitudes in the Tropics. For the purposes of this paper, cows and wetlands could be treated together as a single source category.

Figures – a bit tough to read as so dense... note also the comparison with the NOAA sine-latitude-time plots

Page 8 line 1-4. Note Dlugokencky’s comment that the whole 1980-2007 curve looks just like an equilibration curve – same process throughout.

Page 8 line 9. Ref 4 (Turner et al) was fairly comprehensively shot down by various papers – e.g. Bruhwiler. Better to cite the Rigby paper.

Page 9 line 5 – co-temporaneous? Or coNtemporaneous?

Page 12 lines 1-10. There is a lot of very out of date information here and this whole paragraph needs to be completely rewritten. 58-71% SE Asia is nonsense. The peatland information from Page et al long ago has been heavily added to, including by Page’s own group. See Dargie et al. 2017 Nature 542:86 on the Congo peats, and Xu et al. (2018) PEATMAP, Catena 160:134. Other topics to discuss are rice cultivation (probably hasn’t changed much but warmer) and East and SE Asian cattle and water buffalo populations.

Page 13 line 23....maybe discuss a little more?

Page 14 line 14 Also give evidence for a reduced OH in 1997-8

Incidentally, a general gripe about the word ‘concentration’ – this word comes from bucket chemistry with water solutions. In the air we talk of mole fraction and mixing ratios. For OH in particular, it’s really the lifetime that’s most illustrative, or the oxidising capacity of the atmosphere (and note Cl is involved also).

Page 14 line 11 – I’m a bit sceptical that direct ocean emissions were seriously important. Also the Cl sink should be considered here.

Page 15 line 5 – 6.5 to 8.8 yr residence time seems a bit short – is this OH+Cl+soil??

Page 16 line 5 – see for example the soil uptake work at Sodankyla. Dinsmore et al. Biogeosciences 14(4), pp. 799-815

Page 17 Fig 4 caption bottom line – ‘declined’ ??? rewrite – sounds as if it declined a cigarette.

Page 18 line 1 – concept of multidecadal cycle is introduced – it needs some explanation.

Page 18 line 13 – ‘historical warming hiatus’ - ?????? what does that mean? No hiatus here – it’s been hot!

Responses to reviewers' comments to "Impact of Interannual and Multidecadal Trends on Methane-Climate Feedbacks and Sensitivity"

Thank you and the reviewers for the useful comments on our manuscript. We would like to highlight below major changes in this revision:

- 1) Given the recent release of the IPCC AR6 report, we have updated our introduction and parts of the discussion to include reference the latest relevant literature.
- 2) We have improved our estimates of the climate-driven contributions by using zonal-mean instead of global-mean methane concentration data.
- 3) In addition to C_{CH_4} data, we have incorporated the isotope ratios ($\delta^{13}CH_4$) into our analysis and discussion, as suggested by one reviewer.
- 4) This additional analysis has allowed more detailed hypotheses to be proposed for both the interannual and multidecadal variability of negative-positive feedbacks.
- 5) A comparison of net methane-climate sensitivity with IPCC AR6 is given, and the discrepancy is explained within the context of our proposed hypotheses.
- 6) The rationale of the method is explained in greater detail in the main text and expanded upon further in the Supplementary Information. Further details on methodology are available in a preprint manuscript at <https://gmd.copernicus.org/preprints/gmd-2021-196/> which is itself under revision.

Within our revision, we remain focused on the impact of interannual and multidecadal trends, consistent with the earlier version. Below are the point-by-point responses *highlighted in italic form*.

Reviewer #1	Our response
The authors fit a statistical model based on normalized information flow to estimate the rate of change of CH₄ based on temperature and precipitation. I agree with the authors that a better understanding of changes in methane-climate feedbacks are important and also believe that statistical methods such as information flow, transfer entropy, etc can be very valuable tools in detecting coupling between variables and potential feedbacks, but I am not convinced at this point that the manuscript convincingly elucidates the interplay between positive and negative feedbacks. Presentation of methods and results I think that this manuscript is a case in which presentation of methods, figures, and results gets into the way of understanding the central claims of the paper. For example: - the four included figures present a total of 60(!) subplots, most of which are shaded pseudocolor plots for which it is difficult to discern clear differences.	We tried to reduce the number of total sub-figures in the manuscript by removing or combining some non-critical sub-Figs. However, due to additional analysis on $\delta^{13}CH_4$, the number of Figs and subfigs have increased. To help readers identify the key difference, we have also included non-climate contributions by area-mean in Fig.1, which is the difference between reconstructed observation and area-mean climate contributions. Within the text we are careful to make reference to particular frames within each figure at the appropriate point. Our treatment is data heavy and relies on the comparison of several forms of data, both distributed geographically and temporally, as well as their gradients and deviations. It is difficult to simplify the figure distribution further without taking away from our arguments. We recognise that this paper is both data and description heavy and this alone means that it requires attention and energy to assimilate, but such is the nature of this analysis. Regarding the colour scheme, we have differentiated mainly between red and blue with different darkness to maximize the visible contrast.
- axis labels are sometimes missing and figure headings are small. For example, it is necessary to zoom into Figure 2 to see the difference between the b and c column and column headings don't easily translate from variables/	Fig. 2 has been revised for reader friendliness.

jargon to a physically understandable process.	
- In general the paper uses a lot of shorthand jargon, which makes parsing contents difficult. For example (p9, L8) "Contributions with positive nIFc,SST, positive nIFc,LSAT and negative nIFc,Pr are contemporaneous as are contributions with negative nIFc,SST, negative nIFc,LSAT, and positive nIFc,Pr. This suggests significant indirect SST-feedbacks through the LSAT and Pr" > If I understand correctly, this sentence could be summarized that there is both positive and negative information flow from temperature and precipitation at the same time. I am not sure how the second part of the sentence then follows from that. My general recommendation would be to reduce the amount of variable names and to at least partially translate these concepts into plain language.	We have rephrased this sentence and others throughout the manuscript to improved the reader-friendliness across the manuscript.
- The way the method section is divided between main text and supplement makes it difficult to understand, what exactly is calculated from what. I feel that instead of having eq1 and eq2 in the paper the paper would be better served with providing a clear and concise description of the basic assumptions of the information flow method as applied in this manuscript including some of the basic limitations. The methods section in the supplement should then contain a clear description of the calculations as they are computed.	We have provided a more description about the method, especially the hypothesis within the main text, as suggested. As the method of normalized information flow is not the focus of this paper, more discussions regarding the limitations and caveats around using this method can now also be found in the Supplemental Information.
Local vs non-local effects and validity of results: It is my understanding that the authors use local fields of SST/LandT and Precip together with a boundary layer dataset of CH4 to estimate establish whether changes in T and P inform changes in CH4. My biggest concerns about this as presented here is the following. T and P are local fields, but - given the long (20 or so year) lifetime of methane and the monthly timestep - the change in methane in each grid cell is a composite of local effect and atmospheric transport and atmospheric mixing within a hemisphere occurs on a two-week timescale. This makes me question whether the resulting information flows present a detectable process locally or "just" the result of the statistical model having to fit the nIF locally. The result of that could be that the nIF basically	We agree that differentiating the local vs non-local effects is one of key limitation of our method and hence we have applied different approaches (exclusive land-mean, sea-mean, and area-mean) with different underlying assumptions to estimate the zonal mean of climate-driven contributions. Nevertheless, while zonal mixing takes only weeks, meridional mixing takes months to a year or more. Taking the atmospheric stable CO₂ as an example, at the south pole the CO₂ mixing ratio reached 408 ppm in Jul 2019 but the same level was reached at the equator in Dec 2018, at 30°N in Dec 2017 (12-month mean), and at 60°N in Aug 2017 (12-month mean) (NOAA data). This represents a meridional mixing-time of up to 2 years. The NH-SH mixing-gap is even larger for methane, which could be > 10 years, though this could be partly due to the amplification by faster emissions in the northern hemisphere and faster atmospheric methane oxidation

average to something closer to zero over time, which then may be interpreted to changing feedback. One key piece of information not shown by the author would be the reconstructed field of dCH_4/dt (similar to figure 1a) from their analyses rather than the temporal figures 1m-q, which are not compared to observations (or a comparison of 1m-q to data).	in the southern hemisphere. Since our analysis focuses on meridional distributions and interannual variations with each value of normalized information flow estimated from 49-month-data sets, and the estimated climate contributions reasonably match the observed pattern (with a certain lead time), we believe that the results remain valuable and can provide a better understanding of methane-climate feedbacks. Regarding the last point of this comment, the climate-contributions to the dCH_4/dt, denoted as $\Delta CH_4/\Delta t$, have been shown in subfigures.
-Building on the previous point, it is not clear to me how to verify that information flows are statistically significant. Generally speaking establishing significance in information theory methods is not easy and requires for example bootstrapping. I think that this is critically lacking.	Because we use the magnitude of nIF as a quantifier for the causal contributions, instead of using it as a filter for selecting variable(s) and adopting another statistical method, we use the maximal likelihood value of IF for estimating nIF, without estimating if they are statistically significant. In other words, we use the maximal likelihood nIF to quantify the causal sensitivities and contributions. Furthermore, the statistical significance of IF also depends on the length of the data series and how many trend variations can be used to analyse the causal relation. Since we optimize the analysis for interannual variability and only 49 data points are used for each estimate of IF and nIF, the significance level for the estimated IF and nIF may indeed be low. Having said that, the maximal likelihood still provides the best estimates. In addition, our results are rather consistent with literature findings. Hence, we have not explored further the significance level for IF and nIF.
- the authors apply a single 'calibration' factor (i.e. scaling factor) of estimated results, to estimate the temperature and precipitation driven change in CH₄. It is not apparent to me whether this is justifiable and the choice of this should be clearly justified in the methods.	We have included further discussion on the choice of proportional relationship into the main text and Supplementary Information. In short, the "calibration factor" represents the maximal causal sensitivity.
- To what extent is the assumption valid that patterns in dCH_4/dt represent climate contributions and not for example time varying changes in anthropogenic emission. I am not saying that this is the case, but should be addressed in the manuscript.	We are grateful for this suggestion and have revised related discussions. Instead of assuming anthropogenic emissions fully under non-climate contributions, we provide estimates of non-climate contributions and compare the trends with literature findings of anthropogenic emissions. They are generally consistent.
Diagnosing feedback - It is not clear to me to what extent this is actually diagnosing feedback. Information flow diagnoses directional coupling between variables, but that is not the same as feedback. For example, the paper diagnoses $IF(T \rightarrow CH_4)$, but a true diagnosis of feedback requires diagnosing the entire loop including $IF(CH_4 \rightarrow T)$	We would like to clarify that the sign of nIF_{θ} is adjusted based on correlation. Whether a positive or negative correlation implies positive or negative feedback, requires our understanding of both directions. Since we know that increasing C_{CH_4} will lead to a higher temperature, a positive T-correlation with C_{CH_4} implies positive feedback. However, the case is the opposite for $\delta^{13}CH_4$ since a decrease of $\delta^{13}CH_4$ implies increasing

or some other loop. I don't think that positive and negative nIF values can be assumed to be stand-ins for positive and negative feedback. This should be clearly demonstrated or elaborated on.	biogenic emissions or weakened sink. Hence, a negative correlation will imply positive feedback. Also, due to the uncertain influence of methane on precipitation, when we discuss the precipitation influence on C_{CH_4} and $\delta^{13}CH_4$, we try to avoid using the word "feedback" and instead use the word "correlation".
Reviewer #3	Our response
The manuscripts quantify the contributions of changes in temperatures and precipitations to the atmospheric methane variability based on historical data. Changes in temperatures and precipitations could induce both positive and negative feedbacks through changes in climate driven emission or sinks (e.g., H₂O-OH or CO-OH). Events with different dominant feedbacks are discussed. The combined feedbacks have been shown to limit the climate driven methane emissions and its sensitivity to climate-feedback in the future. The results highlight the significance of climate-driven contributions to the methane variability. The manuscript is well written but the methodology needs additional clarification. The reviewer therefore suggests a minor revision.	Thank you. We have gone through a thorough revision in the light of further comments, and now cover more feedback processes and including elaborated discussions about the method in the Supplementary Information.
General comments: Changes in temperature and precipitation induced impacts on wetland emissions are discussed in the manuscripts. How about the induced impacts on anthropogenic emissions? For example, temperature and precipitation would also affect rice paddies, which is a large source of methane, especially over tropics. It is worthy to include related discussion in the manuscript. On the other hand, the combined contributions from changes in temperatures and precipitations are likely to be biased to some extent in this study, as in many cases, changes in temperatures would affect water cycle and therefore precipitation. Also, it would be more straightforward if we could have a more quantitatively estimates in the land/sst contributions.	We agree that the estimated c-contributions could be influenced by anthropogenic processes too. Vice versa, anthropogenic activities could also be influenced by climate factors. That is why we modify our equations 1 and 2 to better represent this. Nevertheless, we maintain that the major feedbacks are still between natural biogenic processes and climatic factors. Regarding the paddy field emissions, they are much better captured by our analysis on $\delta^{13}CH_4$ as compared to estimates by C_{CH_4} and we have incorporated related discussions. Regarding the comments on bias due to precipitation and the preference to have quantitative estimates of land vs sst contributions, we agree with the reviewer but that may be beyond the limits of our current method. That is why we have made different assumptions for the zonal and global means. Nevertheless, the estimates by the different approaches are rather consistent, while the area-mean provides the best match with the observational trends.
Specific comments: Pate 3, lines 21-24, what about their impacts on rice paddies?	We have incorporated discussions on paddy fields.

Page 4, line 14, should be “as well as positive feedbacks”?	Yes, thank you for pointing out our mistake. Nevertheless, with a major revision in this version, that sentence has subsequently been deleted.
Page 4, line 16, how does this compare to your SST-contribution estimated in this work?	We do not quantify exactly the contributions by ENSO. Instead, we quantify the contribution from each grid (with two different assumptions for the zonal mean). Hence, we are not able to give a precise answer to this comment. Nevertheless, we have incorporated significant relevant discussions about feedbacks with ENSO especially for tropical regions in our revision and the original sentence has been deleted.
Page 5, lines 5-10, how do you define anthropogenic activity here? Do you assume they are not affected by climate variables? But in fact, climate change would affect anthropogenic emissions. Also, how do you determine σ ?	We have revised related equations and discussions to “indirectly” reflect that anthropogenic activity is not necessarily fully independent from climatic factors. The σ is obtained by dividing the c-contributions by the observed trends, as explained now at equation 2, p. 5, l. 12.
Page 7, Figure 1, in Fig1f, if I understand correctly, the observation is based on global mean, the land contribution is based on land mean, and the SST contribution is based on sea mean. So what does the combination of the land contribution and SST contribution suggest here? If we add land-contribution and sea-contribution together, it is much larger than the observations. I do not think anthropogenic emissions have negative contribution here. Similar for Fig.1l. What is a better way to interpret this? On the other hand, is it possible to show the contribution in a more quantitative way, say in %?	There is a misunderstanding here. There are two approaches for estimating the zonal-mean contribution. The first is the exclusive land-mean or exclusive sea-mean, and the second is the area-mean. The global contribution is then the area-weighted sum of zonal-mean contributions from each latitude. We have revised our manuscript to make this clearer. In brief, land-contribution and sea contribution can only be added together when we adopt area-mean,. The % contribution can now be found in Supplementary Table 1.
Page 8, lines 1-5, how do you get this? Figure 1 only shows the climate-drive contribution, which gradually decreases from 1980s to mid2000s. Also, from Eqn (2) for Q_{nc} term, besides the increasing sink, how about the trend in $Q(nat,ant)$? Is it possible to quantitatively compare the contributions from Q_{nc} and Q_C ?	We have revised Fig 1 to incorporate estimated non-climate contributions, which include the change of mean $Q(nat, ant)$ and the negative concentration feedback. The estimated contribution is, however, the combined effect of changing emissions and sinks. Hence, we are unable to provide the estimates of absolute emissions and sinks without a process model. Nevertheless, we have incorporated discussions about positive and negative feedback sensitivity, which are associated with the change in sources and sinks.
Page8, lines6-10, “But these appear to eventually diminish...”, based on what?	We have changed the word “diminish” to “eventually decrease”. With the revised calculation, this trend is less obvious. But the reviewer may refer to Supplementary Table 1.
Page 12, lines 1-3, what about agricultural emissions?	We have incorporated more discussions on climate influence on agricultural emissions especially the paddy field emissions in Asia, based on estimates given by $\delta^{13}CH_4$ trends.
Page 14, lines 1-8, Doesn't it mean due to reduced OH levels from intensive wildfires, it	The revised manuscript includes expanded discussions of the sequential influence of wildfires. To analyse the

tends to increase CH₄ and therefore contributes to a peak in 2020? I do not think a negative SST-H₂O-OH feedback itself could have such big impact on the increase in CH₄. Also, the 1997-1998 ENSO is much stronger than 2019-2020 but the changes in CH₄ are smaller than 2019-2020.	contributions of various feedbacks to the sharp increase of C_{CH₄} in 2020, we will need to have data up to 2022. Nevertheless, if we investigate the meridional distributions, the sharpest rise in 2020 occurred in the tropics and ~ 40°N. These coincide with the severe fires in the western US and Brazil, implying positive feedback contributions. Nevertheless, there should still be some influence through the lowering tropical SST in 2020 (a La Niña year) especially on its influence on rising tropical C_{CH₄}. In our revision, our description on the attribution for the year 2020 is shortened into “The continuous c-contributions via alternating feedbacks explains the strong dC_{CH₄}/dt peak seen from 1997 to 1998 and again in 2020, since both periods experienced intense wildfires followed by a La Niña year.” This does not further differentiate the magnitudes of contributions from positive and negative feedbacks.
Page 15, line 18-19, does this mean methane variability would not be sensitivity to future climate change? In other words, non-climate contribution would play a role in the future?	When we talk about mitigation, it is commonly believed that the non-climate contributions play a greater role than c-contribution, since the former can become negative with reduced anthropogenic emissions. However, with further warming, the interannual and multidecadal methane variability would probably be amplified. We have provided discussions together with Fig. 6.
Page 17, Figure 4, why do you set the same fractional area (i.e., 0.25) for both northern and southern mid/high latitudes? Is it more reasonable to have a higher fractional area in the Northern Hemisphere? Where is table 1? There is no filled red circle in Fig4i.	Sorry for missing out the table in the previous version. We have included it (Table S1) into Supplementary Information. We have also revised the Fig to incorporate the meridional distribution of changing methane-climate sensitivity. By the way, the filled red circle is at (0.58 °C, 16.8 ppb/yr). Note that we change the units of y-axis have changed from Tg/yr in the previous version to ppb/yr in our revised version.
Page18, lines 5-8, can CMIP6 future projections provide some constraints on the estimates here?	We will suggest using our method to constrain Earth System Models, in order to identify their potential shortcomings regarding their capability of estimating source and strength of feedbacks. Nevertheless, we feel this is beyond the scope of this paper as it would require extensive further discussion.
Supplement: Page5, line9, should be “SST-driven”	Thank you. We have rewritten this paragraph.
Page5, lines 20-22, does this mean the sum could lead to overestimates in the overall territorial contribution?	Yes. We have also highlighted the potential overestimates of $\partial C_{CH_4, T\&P_r} / \partial t$ due to the method limitation. But it is unlikely to affect our key findings in this manuscript about the impact of interannual / interdecadal climate variability.
Reviewer #4 (Remarks to the Author):	Our response
General This is a very interesting pioneer paper. I’m not sure I agree with some of its suppositions, and I	Thank you for the very useful comments. We have incorporated them and explored more analyses.

think some of the inputs are either dated or wrong, but the paper is very innovative and definitely interesting, and potentially offers a major new route to solving an important question. Thus I support publication after some revision. Arguably the most interesting question in studying atmospheric methane is not why it is rising at record growth rate right now (which is an extremely interesting puzzle), but the even more important puzzle whether the warming is feeding the warming. We all have strong suspicions that warming is indeed driving methane emission and hence driving warming, but these are largely gut feelings: it is very tough to say this specific emission rise comes from this specific temperature rise. This paper addresses that specific problem – using Liang’s ‘normalised information flow’ approach, it develops a methodology to find causal connections.	
Now I need to caution that casual coincidence is NOT proof of causal connection, but it’s interesting, really interesting.....	We have incorporated more discussion about the method in the main text and SI.
That said, I have a number of specific quibbles with the paper. In particular, the peatland information is very badly out of date (page 12) and the lack of mention of the tropical ruminant source and the Cl sink both need to be considered.	We have revised the manuscript to incorporate relevant information into the discussions. For example, we have cited the most recent soil organic carbon map (not just peat map), and many other papers discussed in the recent IPCC AR6 report.
Note, there is barely any use of isotopic information.	We agree that the use of isotopic information is very important and have incorporated them into this revision
Thus I recommend publication after revision.	Thanks.
Specific Please could Nature Comms send out its review documents with Line Numbers continuously numbered through the text pages....it is so much easier on the referee and so easy to do!!	We have added continuous Line Numbers in the revision.
Abstract: line 6-7 “SST related OH likely” – rewrite as the English is ugly and ambiguous.	Abstract Rewritten
Page 4 line 7-9. No mention here of the Cl sink. I know it is small but it is significant when discussing ocean emissions, and it is latitudinally distributed. It should be included. Various papers on this - see for example Hossaini, R., et al. (2016) A global model of tropospheric chlorine chemistry: Organic versus inorganic sources and impact on methane oxidation. Journal of Geophysical Research: Atmospheres 121.23 (2016).	Noted with thanks. We have incorporated it into the revised manuscript.
Page 4 line 14. The problem of ENSO and wetlands is not simple. First, both heat and	Thanks for the comment. With the revised estimates based on concentration data

water are involved. Because methane emission is exponential with Temperature – Arrhenius relation – then a hot but fairly dry wetland can in principle emit more methane than a cool wetland. Secondly there is time factor – groundwater is important. If the previous season was wet, then a dry El Nino season may still produce enough run off to produce a soggy wetland, while conversely during a wet La Nina year the run off may be so consumed in the task of raising groundwater levels that there is not much expansion of wetland area.	at each latitude, our results suggest positive contributions with negative feedback with ENSO index are often more significant than positive feedbacks for tropical areas. This is also consistent with the cited literature. Nevertheless, with future warming and possibly amplified variability, we cannot be sure whether the situation will change.
Page 4 line 17 – this is the only mention of isotopes, yet surely isotopes are central to solving the anthropogenic vs natural puzzle! That’s arguably the big weakness of the whole paper.	Thank you. We have revised the manuscript and incorporated analysis with isotope data. The results do indeed provide useful insights into the trends of climate and non-climate contributions.
Page 5 line 12 – Normalised information flow. I think this concept needs an introductory paragraph. Yes, readers can read Liang’s papers, but it would help to introduce the concept in the text here.	We have incorporated a brief description of the hypothesis behind the method in the main text and SI.
Page 6 lines 2 and 3. In this context it might be worth mentioning the NOAA Sine-Latitude-Time plots....see Growth Plot in: https://gml.noaa.gov/ccgg/figures/ The plot is also used in the Nisbet et al. paper.	We were referring to the reconstruction of climate-driven contributions, not just observational data from NOAA. Nevertheless, we have removed this sentence.
Page 6 line 20. Careful – wetlands ARE important but so also are tropical cows (see Schaefer et al 2016, and both Nisbet et al papers). Moreover cows and wetlands are very similar – similar methanogens in the grass-to-methane factory, same link to rainfall (more rain, more fat cows), same isotopic signatures, and similar latitudes in the Tropics. For the purposes of this paper, cows and wetlands could be treated together as a single source category.	From the results based on isotope data, tropical/subtropical paddy fields also contribute significantly to the climate-contributions. The soil sink at nearby uplands to these paddy fields are also influenced significantly. However, we postulate that emissions from cows probably do not depend on climate significantly and therefore they are not discussed..
Figures – a bit tough to read as so dense... note also the comparison with the NOAA sine-latitude-time plots	We have attempted to improve the readability of the Figures. We acknowledge that our paper is data and analysis dense but such is the nature of this approach. We have also attempted to improve the readability of the text in this version.
Page 8 line 1-4. Note Dlugokencky’s comment that the whole 1980-2007 curve looks just like an equilibration curve – same process throughout.	Noted. We have further extended the discussions and hypothesized a multidecadal oscillation between concentration- and climate-dominant sinks.
Page 8 line 9. Ref 4 (Turner et al) was fairly comprehensively shot down by various papers –	Thanks. We have revised our reference.

e.g. Bruhwiler. Better to cite the Rigby paper.	
Page 9 line 5 – co-temporaneous? Or coNtemporaneous?	Thanks. Revised.
Page 12 lines 1-10. There is a lot of very out of date information here and this whole paragraph needs to be completely rewritten. 58-71% SE Asia is nonsense. The peatland information from Page et al long ago has been heavily added to, including by Page’s own group. See Dargie et al. 2017 Nature 542:86 on the Congo peats, and Xu et al. (2018) PEATMAP, Catena 160:134. Other topics to discuss are rice cultivation (probably hasn’t changed much but warmer) and East and SE Asian cattle and water buffalo populations.	Thank you for pointing out our outdated information. We have revised accordingly. However, we still do not cover the impact of cattle and cows, since we are not aware of their confirmed link with climatic factors, and the identified areas from isotope analysis appear to match the paddy field areas better.
Page 13 line 23....maybe discuss a little more?	We have revised and incorporated more discussions about the influence of ENSO on tropical C_{CH4} trends.
Page 14 line 14 Also give evidence for a reduced OH in 1997-8	This earlier page and line number are about the potentially underestimated direct oceanic emissions driven by SST warming. We have revised the relevant discussions.
Incidentally, a general gripe about the word ‘concentration’ – this word comes from bucket chemistry with water solutions. In the air we talk of mole fraction and mixing ratios. For OH in particular, it’s really the lifetime that’s most illustrative, or the oxidising capacity of the atmosphere (and note Cl is involved also).	We note the suggested use of mixing ratio over concentration for the gas mixture. Nevertheless, it is much more common for climate change researchers to use “concentration” than “mixing ratio”, as seen in the IPCC AR6.
Page 14 line 11 – I’m a bit sceptical that direct ocean emisissions were seriously important. Also the Cl sink should be considered here.	Yes, we understand this scepticism. Nevertheless, in this revision, together with the analysis based on isotope data and remap of the SST-contributions on a coordinate of SST x time, we are even more certain about the role of direct oceanic emissions and provided more discussion of it. Having said that, we are of course open to other possible mechanisms that can explain our results.
Page 15 line 5 – 6.5 to 8.8 yr residence time seems a bit short – is this OH+Cl+soil??	Yes, that considers all sinks, including OH, Cl, and soil. We have also incorporated the number from the IPCC AR6.
Page 16 line 5 – see for example the soil uptake work at Sodankyla. Dinsmore et al. Biogeosciences 14(4), pp. 799-815	Noted. But we have cut down the discussion about soil sink in this section. Beyond the scope of this paper, we may like to suggest more ground investigation to the Eastern Russia uplands due to the strong detected signals (Figs. 1u, w, 4j and 5f).
Page 17 Fig 4 caption bottom line – ‘declined’ ??? rewrite – sounds as if it declined a cigarette.	Revised. Thanks.
Page 18 line 1 – concept of multidecadal cycle is introduced – it needs some explanation.	Thanks. We have revised and incorporated two hypotheses behind the multidecadal cycle.
Page 18 line 13 –‘historical warming hiatus’ - ?????? what does that mean? No hiatus here – it’s been hot!	We have rewritten this sentence together with a revised last paragraph.

REVIEWERS' COMMENTS

Reviewer #1 (Remarks to the Author):

I would like to thank the authors for the revisions to the manuscript, which I believe have been beneficial for clarity and contents.

With respect to my general comments about assumption of locality in the information flow. I appreciate the authors comments and think that some of this discussion should be mentioned in the paper as a limitation of the method.

The same is true regarding the statistical significance of results. I also noticed that the authors use significant for example on line 142, but mean considerable. I suggest to remove the word significant from the paper as there is no way to ascertain patterns are indeed statistically significant.

I still believe that there are several areas, in which readability of the manuscript could be strengthened. For example, Figure 2 uses +ve and -ve as shorthand in figure titles, but these are not used or explained in the text at all.

Similarly, Figure 6 e has a large assortment of different curves, for which it is not clear to me how these were derived from the the data. I also looked into the methods section, but could not find this. Given that the authors claim a high climate sensitivity of methane that exceeds IPCC consensus, this should be more carefully explained and reasoned.

Point-by-point responses (in *bold italic*) to the reviewer:

Reviewer #1 (Remarks to the Author):

I would like to thank the authors for the revisions to the manuscript, which I believe have been beneficial for clarity and contents.

With respect to my general comments about assumption of locality in the information flow. I appreciate the authors comments and think that some of this discussion should be mentioned in the paper as a limitation of the method.

The same is true regarding the statistical significance of results. I also noticed that the authors use significant for example on line 142, but mean considerable. I suggest to remove the word significant from the paper as there is no way to ascertain patterns are indeed statistically significant.

Responses: Thank you for the comments, we have revised the Methods section and included relevant discussions under a section within the methods termed "Caveat". We have removed the term "significant" from the text where it could be mistaken for a statistical term rather than simply "considerable" etc.

I still believe that there are several areas, in which readability of the manuscript could be strengthened. For example, Figure 2 uses +ve and -ve as shorthand in figure titles, but these are not used or explained in the text at all.

Responses: Thank you for the suggestions, we have inserted these abbreviations in the main text as well as revised some Figure captions.

Similarly, Figure 6 e has a large assortment of different curves, for which it is not clear to me how these were derived from the the data. I also looked into the methods section, but could not find this. Given that the authors claim a high climate sensitivity of methane that exceeds IPCC consensus, this should be more carefully explained and reasoned.

Responses: Thank you for the comments, we have revised the Methods section and included another sub-section for Fig. 6e.